# Nanofiber Scaffolds as Drug Delivery Systems Promoting Wound Healing

**DOI:** 10.3390/pharmaceutics15071829

**Published:** 2023-06-26

**Authors:** Ziwei Jiang, Zijun Zheng, Shengxiang Yu, Yanbin Gao, Jun Ma, Lei Huang, Lei Yang

**Affiliations:** Department of Burns, Nanfang Hospital, Southern Medical University, Jingxi Street, Baiyun District, Guangzhou 510515, China; jiangzw2025@163.com (Z.J.); zhengzj995@163.com (Z.Z.); keyanysx@163.com (S.Y.); gaoyanbin6@yeah.net (Y.G.); nfyy_majun@163.com (J.M.); mrbonejuste@sina.cn (L.H.)

**Keywords:** wound healing, nanofiber scaffold, polymer, drug delivery, stimulus response

## Abstract

Nanofiber scaffolds have emerged as a revolutionary drug delivery platform for promoting wound healing, due to their unique properties, including high surface area, interconnected porosity, excellent breathability, and moisture absorption, as well as their spatial structure which mimics the extracellular matrix. However, the use of nanofibers to achieve controlled drug loading and release still presents many challenges, with ongoing research still exploring how to load drugs onto nanofiber scaffolds without loss of activity and how to control their release in a specific spatiotemporal manner. This comprehensive study systematically reviews the applications and recent advances related to drug-laden nanofiber scaffolds for skin-wound management. First, we introduce commonly used methods for nanofiber preparation, including electrostatic spinning, sol–gel, molecular self-assembly, thermally induced phase separation, and 3D-printing techniques. Next, we summarize the polymers used in the preparation of nanofibers and drug delivery methods utilizing nanofiber scaffolds. We then review the application of drug-loaded nanofiber scaffolds for wound healing, considering the different stages of wound healing in which the drug acts. Finally, we briefly describe stimulus-responsive drug delivery schemes for nanofiber scaffolds, as well as other exciting drug delivery systems.

## 1. Introduction

Cutaneous tissue traumas, such as burns, injuries, surgical incisions, limb ischemia, and chronic ulcers, are common surgical problems [1]. Skin tissue transplantation, as a traditional treatment method, cannot provide certain efficacy due to issues related to survival rates [2]. Additionally, when the wound area is large, surgery often needs to be completed multiple times, imposing significant economic, physiological, and psychological burdens on the patient [3]. Furthermore, antibiotics given orally or by injection cannot guarantee a sufficient drug concentration at the wound site and may cause side effects in other systems [4]. Traditional gauze dressings combined with drugs do not possess gas–liquid exchange ability and good moisturizing properties. Additionally, gauze, when used as a drug carrier, cannot achieve stable and sufficient drug release. Therefore, the development of a multifunctional dermal substitute that can be applied to the wound bed is an urgent requirement in the clinical context [5]. An ideal wound dressing should have good biocompatibility, degradability, breathability, water absorption, dust resistance, and antibacterial properties, and should protect the wound from harmful external mechanical stimuli [6]. Furthermore, it should be easy to remove from the wound without causing tissue damage. At the same time, the spatial structure of the wound dressing should allow for the maximization of cell interactions, thus facilitating cell proliferation and migration [7].

Considering their potential to meet these requirements, multifunctional bioactive wound dressings have become a research hotspot [8]. In this context, nanofibers are excellent candidates for engineering skin-tissue scaffold materials and can be used to construct complex drug delivery systems [9]. Nanofibers possess a natural extracellular matrix (ECM) structure, high water absorption, high interconnected porosity, breathability, and moisture permeability, creating a suitable environment for hemostasis, prevention of exogenous infections, cell migration and proliferation, cell respiration, and exudate absorption. The ideal microstructure of nanofibers mimics the structure of the ECM, promoting cell adhesion, growth, and proliferation [10]. Due to their porous network structure, nanofibers make it easy for oxygen to diffuse to the wound location. At the same time, their good moisture retention ability allows nanofiber dressings to provide a moist environment for the wound, making them less likely to adhere to damaged surfaces. Compared with block materials such as hydrogels, the high porosity and larger surface area of nanofibers have beneficial effects on cell activity [11]. Furthermore, the unique physicochemical properties of nanofibers make them an ideal carrier for drug delivery [12]. By delivering biologically active substances such as growth factors, cytokines, and anti-inflammatory and antibacterial drugs, drug-loaded nanofiber scaffolds can promote hemostasis [13]; reduce chronic inflammation [14]; promote the proliferation, migration, and secretion of cells; help to form new blood vessels [15]; inhibit scar formation [16]; and fight against bacterial infections, thus promoting wound healing [17,18].

Many polymers can be used to prepare nanofibers, mainly divided into natural and synthetic polymers. Some interesting carbon-based materials have also recently been reported; for example, bioactive graphene quantum dots (GQDs) are one of the newest carbon-based nanomaterials with antibacterial and antidiabetic potential. Due to π–π aggregation and hydrophobic interactions with highly specific surface properties, GQD-based composites have a high drug-carrying capacity, making GQDs an excellent candidate for nanomaterials [19]. Electrostatic spinning is the most commonly used technological means for preparing nanofibers, which can be loaded with a variety of bioactive agents for the promotion of wound healing. However, the choice of electrospinning feedstock is a question worth exploring and many natural polymers with excellent properties are not spinnable, such as chitosan. Additionally, how to load a drug onto a scaffold without losing its biological activity is an important technical issue in the preparation of electrospun drug-loaded scaffolds [20]. Furthermore, the choice of raw material(s) for the preparation of nanofibers is an important issue, and the balance between mechanical and biological properties is a key challenge that researchers have been trying to overcome. In addition to mechanical and biological properties, some polymers have additional interesting features, such as intrinsically conducting polymers (ICPs), represented by polythiophene (PTh), polyaniline (PANI), and polypyrrole (PPy). For example, PTh, in addition to its unique electrical behavior, possesses excellent properties such as high environmental behavior, thermal stability, and mechanical robustness, allowing for the synthesis of nanocomposites with excellent properties to manage the process of releasing drugs from the nanofibers [21].

This paper aims to review the research progress and applications of drug-loaded nanofiber scaffolds to promote healing from trauma in skin tissues. First, we introduce commonly used methods for nanofiber preparation, including electrostatic spinning, sol–gel, molecular self-assembly, thermally induced phase separation, and 3D printing techniques. Next, we summarize the polymers used to prepare nanofibers and the drug delivery methods for nanofiber scaffolds. We then review the application of drug-loaded nanofiber scaffolds in wound healing in terms of the different stages of wound healing in which the drugs act. Finally, we briefly describe stimulus-responsive drug delivery schemes for nanofiber scaffolds, as well as discuss other exciting drug delivery systems.

## 2. Nanofiber Scaffold Technology

Various methods for the preparation of nanofibrous scaffolds have been described in the literature [22] with common preparation methods mainly utilizing electrospinning technology, sol–gel methods, molecular self-assembly methods, and thermally induced phase-separation technology (Figure 1). This paper mainly focuses on and provides an overview of such commonly used nanofiber preparation methods.

### 2.1. Methods for Preparation of Nanofibrous Scaffolds

#### 2.1.1. Electrospinning

Electrospinning is an efficient method for the preparation of nanofibers in medical applications. This technique utilizes a high-voltage electrostatic field to transform polymer solutions or melts into fibrous materials, forming nanoscale fibers. The advantages of electrospinning include its simplicity, low cost, scalability, and the production of fibers with a high surface area and porosity. Consequently, electrospinning technology has been extensively applied in biomedicine, textiles, filtration, and other fields. The electrospinning process mainly comprises several steps: the preparation of polymer solutions or melts, the application of an electrostatic field, the ejection of solutions or melts, the evaporation or solidification of solvents, and the collection of nanofibers. First, an appropriate polymer material is selected and dissolved (or melted) to obtain the desired concentration of a solution (or melt). A high-voltage electrostatic field is applied between the polymer solution or melt and a collector, creating an electric-field gradient. Influenced by the electrostatic field, the polymer solution or melt is ejected from a nozzle to form fibrous materials. During the flight of the fibers, the solvent evaporates or the melt solidifies, resulting in nanoscale fibers. Finally, the nanofibers are deposited on a collector, forming a nanofiber membrane. Parameters in the electrospinning process, such as voltage, the distance between the nozzle and collector, solution concentration, and nozzle diameter, can impact the diameter, morphology, and properties of the resultant nanofibers. By adjusting these parameters, nanofibers with different sizes and properties can be prepared [28,29,30,31,32,33]. As shown in Figure 1a, Jian Li et al. prepared a poly(3-hydroxybutyrate-4-hydroxybutyrate) (P34HB) fibrous wound dressing incorporating antibacterial ciprofloxacin (CIP) and proangiogenic dimethylsalicylic acid (DMOG) using an electrospinning technique [23].

The electrostatic spinning method enables the direct and continuous production of nanofibers with high specific surface area, high porosity, controlled size, and easy surface functionalization; however, the strength of the nanofibers produced by electrostatic spinning is low and the yield is low. The raw material for electrostatic spinning is limited to spinnable polymers and the inability to obtain separated nanofiber filaments or staple fibers also limits the use of this technology.

#### 2.1.2. Sol–Gel Method

The sol–gel method can be employed to prepare medical-grade nanofiber scaffolds for various biomedical applications, such as tissue engineering, drug delivery, and wound healing. These scaffolds can comprise inorganic materials, organic–inorganic hybrids, or bioactive glasses, offering unique properties such as biocompatibility, bioactivity, and controlled degradation. The general process of preparing medical nanofibrous scaffolds using the sol–gel method includes preparing the sol, incorporating bioactive agents, gelation, and drying. Suitable precursors, such as metal alkoxides or organometallic compounds, must be capable of being hydrolyzed and condensed to form the desired material. The precursor is dissolved in a solvent and a catalyst may be added to facilitate the hydrolysis and condensation reactions. If the scaffold is intended for drug delivery or other therapeutic applications, bioactive molecules (e.g., drugs, growth factors, or proteins) can be incorporated into the sol at this stage. The suspension of solid particles is the sol, the solid particles are connected to form a network, and the solvent is removed by freeze-drying after gelation. This step can cause the gel to shrink and densify and may require carefully controlling the temperature and humidity to prevent cracking or other defects. The sol–gel method allows for precise control of the composition, structure, and morphology of medical nanofiber scaffolds. This enables one to tailor their properties, such as mechanical strength, porosity, and degradation rate, to specific biomedical applications. However, the method can be complex and may require the optimization of various parameters (e.g., reaction conditions and drying conditions) to achieve the desired scaffold characteristics [34,35,36,37,38,39,40]. As shown in Figure 1b, Wei Ma et al. used chitosan nanofibers (CSNF), prepared using the pulping homogenization method combined with the sol–gel method, and further introduction of polyvinyl alcohol (PVA), to prepare chitosan nanofiber/nanosilica and chitosan nanofiber/nanosilica/PVA scaffolds [24].

The main advantage of the sol–gel method is the good homogeneity of the raw material at the molecular level; however, a long time (days or even weeks) is typically required for the whole sol–gel process and the nanofibers prepared by the sol–gel method during drying may present high shrinkage due to the large number of micropores in the gel and the escape of gases and organic matter during drying. The main limitations of this method for clinical applications are the complexity of the process, the difficulty of mass production, and the fact that deviations in any of the steps involved may affect the quality and properties of the material due to the need to control several parameters throughout the process (e.g., temperature, pH value, and reaction time).

#### 2.1.3. Molecular Self-Assembly Technology

Molecular self-assembly technology denotes methods involving the organization and arrangement of molecules spontaneously at the molecular scale, in order to form ordered and complex supramolecular structures. This technology has been applied in various fields, such as tissue engineering [41], drug delivery systems [42], optoelectronic devices [43], and sensors [44]. The key to molecular self-assembly is the noncovalent interactions between molecules, such as van der Waals forces, hydrogen bonds, electrostatic interactions, metal coordination, and hydrophobic interactions. Self-assembly enables precise control over the material structure and properties at multiple scales (from nanometers to micrometers). Under appropriate conditions, these interactions guide molecules to spontaneously organize into ordered structures. The self-assembly process can occur in the solution, gas phase, or at solid interfaces, and can be influenced by adjusting the experimental conditions (e.g., concentration, temperature, pH, and so on). The steps for preparing nanofiber scaffolds through molecular self-assembly vary, depending on the materials and methods used. Micelle templating is generally used as the underlying self-assembly strategy [45]. First, appropriate polymers or other molecules must be chosen as the basic building blocks for the construction of nanofiber scaffolds. These molecules should have properties that allow them to self-assemble into fibrous structures, such as biomacromolecules (e.g., peptides and proteins) or synthetic polymers (e.g., polyelectrolytes and amphiphilic molecules). The selected molecules are dissolved in a suitable solvent, forming a uniform distribution of molecules in the solution. As the molecular concentration increases, the molecules self-assemble to form nanoscale micelles (i.e., spherical or rod-like aggregates). Micelles are amphiphilic structures consisting of a hydrophobic core and a hydrophilic outer layer, the formation of which is driven by hydrophobic interactions and van der Waals forces. Under appropriate conditions (e.g., temperature, pH, and ion concentration), the micelles further self-assemble into ordered nanofiber structures. This structural formation is typically driven by the interactions between micelles (e.g., electrostatic interactions, hydrogen bonds, and metal coordination). After self-assembly of the nanofibrous structures, they can be fixed onto a scaffold using solidification methods (e.g., solvent evaporation, cross-linking, and chemical reduction) [46]. The solvent and unreacted materials are removed by washing, resulting in a dry nanofiber scaffold [47]. As shown in Figure 1c, Yosuke Hisamatsu et al. reported a self-assembled amphiphilic 4-aminoquinoline (4-AQ)-tetraphenylethylene (TPE) conjugate using a self-assembly process involving the formation of spherical nanoparticles, transition to short nanofibers, and growth to long nanofibers at room temperature. The time scale of the self-assembly process varies according to the pH reactivity of the 4-AQ molecules in the weakly acidic-to-neutral pH range [25].

The main advantages of molecular self-assembly techniques are that the assembled structures are at the molecular scale—much smaller than the size of structures that can be achieved by other nanofiber processing methods—and they are less costly as the molecular self-assembly process is automatic and spontaneous. Making the self-assembly process more controllable is the desired goal of researchers, and how to control the start and end of the assembly and to observe and control the assembly process in situ are the main technical issues limiting its application.

#### 2.1.4. Thermally Induced Phase Separation (TIPS)

Thermally induced phase separation (TIPS) is a process technique for the fabrication of porous materials, such as polymer foams, fibers, and membranes [48]. This technique is commonly employed for the preparation of microporous polymer membranes, which have extensive applications in filtration, water treatment, biomedicine, and other fields. The fundamental principle of the TIPS technique involves mixing a polymer with one or multiple compatible solvents, followed by adjusting the temperature to achieve phase separation [49]. During the phase separation process, the polymer and solvent separate into two distinct phases: a polymer-rich phase (hard phase) and a solvent-rich phase (soft phase). Subsequently, the solvent is removed from the separated phases through cooling, solidification, and drying steps in order to form a porous polymer structure. The TIPS technique offers certain advantages, such as simplicity, lower cost, and good controllability [50,51]. By adjusting the preparation conditions (e.g., temperature, solvent type, and concentration), control over the porous structure can be achieved, thereby tuning the material properties [52]; however, this method has some limitations, such as the potential for residual solvent production, necessitating additional measures for treatment. As shown in Figure 1d, poly (Llactide)/hydroxyapatite (PLLA/HA) was stirred in a magnetic stirrer for 24 h, sodium chloride(NaCl) was added, and the samples were kept in a freezer for a further 24 h. The frozen samples were freeze-dried at −80 °C by vacuum at ~10–20 Pa for 24 h. The dried samples were removed and placed in desalting water to leach the NaCl out by slow stirring on a magnetic stirrer. After leaching, the stents were dried using an air dryer (60 °C) for 24 h [26].

TIPS technology produces nanofiber scaffolds with small and uniformly distributed pore sizes, high accuracy in terms of pore control, a large adjustable range, high flux, and high mechanical strength and resistance to filament breakage. However, this method has certain limitations, such as fast liquid–solid phase separation, the formation of a skin on the surface of the nanofiber membrane (which affects the flux), and the hydrophobic nature of the resulting membrane (which can easily adsorb proteins and other substances and be contaminated, thus reducing its flux).

#### 2.1.5. 3D Printing Technology

Three-dimensional bioprinting is an emerging technology with a promising future in the fields of tissue engineering and regenerative medicine; 3D printing is considered a powerful tool for developing complex geometries suitable for a wide range of materials [53]. Syringe extrusion printing is the most common 3D printing technology used for pharmaceutical purposes. The materials used in syringe extrusion 3D printing are known as bioinks, including materials such as alginate, gelatin, and so on [54]. The 3D printing technology enables the fabrication of porous structures for tissue engineering and the application of freeze-drying technology in the 3D printing of water-based bioinks allows the material to retain its designed shape and nanoporous structure [55]. The preparation of nanofibers using 3D printing technology first requires the selection of suitable raw materials for the selection preparation of bioinks. For example, R. Olmos-Juste et al. have developed alginate–cellulose nanofiber (ACNF) formulations as suitable bioinks for 3D printing technology. After evaluating their printability and shape fidelity, curcumin was loaded into the bioinks, which were 3D printed and freeze-dried to obtain porous nanofibrous scaffolds [27] (Figure 1e).

The advantage of 3D printing technology is that it has the refinement and repeatability associated with computer modeling; however, the disadvantage is that the size of the prepared fibers is typically large and, although the macroscopic morphology can be controlled, the actual internal nanofiber structure is largely dependent on the properties of the raw material itself (e.g., when printing with cellulose fibers). Furthermore, the high cost of 3D printers and the lack of diversity in the range of bioinks limit the clinical application of this technology at present.

### 2.2. Polymers Used to Make Nanofibers

Nanofibers used for wound dressings or implantable scaffolds need to possess fundamental characteristics such as biocompatibility, biodegradability, and low cytotoxicity for both them and their degradation products, which must be provided by the polymers used to manufacture the nanofibers [56,57]. At present, there are three main types of polymers that can be used to produce nanofibers: natural polymers, synthetic polymers, and hybrid polymers. Natural polymers, such as various proteins and polysaccharides, resemble the components of the natural extracellular matrix (ECM) and possess inherent structural and biological properties [58]. They exhibit significant advantages in terms of biocompatibility, promotion of cell adhesion and proliferation, enhancement of epithelialization and vascularization, low cytotoxicity, biodegradability, and even hemostatic and antibacterial properties. Synthetic polymers, such as polylactic acid (PLA), polyglycolic acid (PGA), polyethylene glycol (PEG), polyvinyl alcohol (PVA), and polycaprolactone (PCL), have advantages in providing finer fiber diameters, higher porosity, and tunable mechanical strength for scaffolds [11]. However, they lack certain biological properties and have issues regarding the toxicity of degradation products and the mismatch between degradation rates and wound-healing requirements. Therefore, in order to develop nanofiber materials that better meet the needs of wound healing, researchers have created hybrid polymers by combining natural and synthetic polymers, modifying the materials to complement their strengths and weaknesses, optimizing the physicochemical and biological properties of the fibers, and reducing their toxicity [59].

#### 2.2.1. Natural Polymers

Natural polymers suitable for the preparation of nanofibers mainly fall into two categories: polysaccharides and proteins. Polysaccharides include chitosan [60], starch [61], alginates [62], hyaluronic acid [63], and cellulose [64]; while protein-based polymers include collagen [65], gelatin [66], fibrinogen [67], silk proteins [68], sericin [69], elastin [70], keratin [71], and plant proteins [72]. Table 1 lists the polymers that can be used to prepare nanofibrous scaffolds.

(a)Chitosan

Chitosan is an alkaline natural polysaccharide substance; in particular, it is a large molecular viscous polysaccharide formed by the partial deacetylation of chitin [115]. The amino groups on the chitosan molecular chain are easily protonated and carry a positive charge, endowing chitosan with a wide range of antibacterial properties and hemostatic effects [116]. The protonated amino groups (NH_3_^+^) of chitosan interact with negatively charged bacterial cell membranes [117]. A polymer film forms on the cell surface, altering the permeability of the cell membrane and disrupting the normal metabolism of the bacteria, thereby inhibiting bacterial growth [118]. Simultaneously, the positive charge carried by chitosan promotes the aggregation of negatively charged red blood cells, increasing platelet adhesion and promoting blood coagulation [119]. In addition to its antibacterial and hemostatic properties, chitosan also possesses some basic characteristics which are ideal for wound dressings, such as biocompatibility, biodegradability, and low toxicity [120]. However, chitosan solutions themselves have high viscosity and are not easily electrospun. Therefore, synthetic polymers are often added to improve their spinnability [121].

(b)Starch

Starch is primarily composed of dehydrated glucose units that form two distinct polymers—amylose and amylopectin—which can be naturally biodegraded [122]. The swelling capacity and water retention ability of starch enable it to absorb plasma, thus promoting blood coagulation. Surface contact between blood components and starch fiber sponges initiates clotting through platelet activation and activation of the intrinsic pathway, influencing the endogenous coagulation pathway in a dose-dependent manner [123]. Additionally, starch can serve as a natural adhesive, inducing the reversible contraction of epithelial cells and promoting the repair of damaged cells [124]. However, the high hydrophilicity and poor mechanical properties of starch hinder its application in wound healing [84].

(c)Alginate

Alginate is a natural anionic polysaccharide derived from brown algae [125]. Calcium ions in the alginate interact with sodium ions in the wound exudate, causing the alginate to form a gel at the site of the wound, providing an appropriately moist environment for wound healing [126]. However, due to the low molecular weight of sodium alginate, it tends to form a gel at slightly higher concentrations, making it difficult to synthesize pure sodium alginate nanofiber materials [127].

(d)Hyaluronic acid

Hyaluronic acid (HA) is an anionic glycosaminoglycan that is widely distributed in human connective tissues, eyes, and skin [128]. It is a major component of the extracellular matrix and plays a role in regulating inflammatory responses during wound healing, promoting the proliferation and migration of keratinocytes and fibroblasts, enhancing collagen deposition and angiogenesis at the wound site, and reducing scar formation [129,130,131]. Although it possesses excellent biological properties, the limited mechanical performance of hyaluronic acid nanofibers restricts its application in tissue engineering [132,133].

(e)Cellulose

Cellulose is a linear form of glucose (also known as glucan), which is naturally synthesized by plants, bacteria, and algae [134]. Cellulose is connected through β-1,4 linkages between glucose residues and can be classified into cellulose ethers, cellulose fibers, cellulose pulp, microcrystalline cellulose, and other similar derivatives [135,136]. Due to its good biocompatibility with human cells, cellulose is widely used throughout the biomedical sciences. The cellulose used in tissue engineering can be mainly divided into plant cellulose (PC) and bacterial cellulose (BC). PC contains impurities such as pectin, lignin, and hemicellulose, has a moderate water retention capacity (60%), and exhibits moderate crystallinity and tensile strength [137,138]. On the other hand, BC is chemically pure, with high water retention capacity (100%) and hydrophilicity, as well as high crystallinity and tensile strength [90]. Compared to BC, PC is more difficult to degrade. When treated with an acidic solution, BC degrades quickly. In addition, phosphate-buffered saline (PBS) may potentially degrade BC without dispersing the existing fiber structure [139]. When applied to tissue engineering, cellulose can be modified to form hydroxypropyl methylcellulose, hydroxyethyl cellulose, and sodium carboxymethyl cellulose to improve its biocompatibility, solubility, stability, adhesiveness, and controllable structural characteristics [90].

(f)Collagen

Collagen is a protein primarily found in animals, particularly in the skin, tendons, bones, and teeth. It has a triple helix structure, formed by three polypeptide chains intertwined through hydrogen bonds. Collagen possesses a certain strength and toughness, providing good mechanical properties for nanofiber scaffolds [140]. As a protein that is widely present in animals, collagen has excellent biocompatibility. This means that collagen nanofiber scaffolds are less likely to cause immune reactions or inflammation when implanted in the body. At the same time, collagen has good cell adhesion properties, promoting cell growth and proliferation on the scaffold surface. Collagen nanofiber scaffolds can gradually degrade in the body and integrate with newly formed tissue. This is beneficial in reducing complications such as the displacement and rejection of implants [141].

(g)Silk fibroin (SF)

Silk fibroin (SF) is extracted from the secretions of the domestic silkworm and is a natural amphiphilic copolymer with both hydrophobic and hydrophilic segments. It has several advantages, including biocompatibility, proteolytic degradation, toughness, thermal stability, and tensile strength. Silk fibroin exhibits good cell adhesion, promoting cell growth and differentiation on the scaffold surface. However, due to its natural origin, the properties of silk fibroin may vary, depending on the source and extraction process. Furthermore, the extraction and purification of silk fibroin are relatively complex, which may increase the cost and difficulty of preparing nanofiber scaffolds [142].

(h)Fibroinogen

Fibroinogen refers to the precursor of fibroin that has not been folded and assembled, which is a water-soluble protein mainly found in silk produced by silkworms [143]. Fibroinogen exists in a water-soluble state in the silk-gland vesicles. When the silkworm begins to spin silk, fibroinogen is excreted through the silk-gland duct and gradually loses moisture. During this process, fibroinogen molecules undergo structural changes, forming strictly arranged β-sheet structures and assembling them into stable fibrous substances. This fibrous substance is a well-known silk, and its main component is fibroin [144].

Fibroin and sericin are both proteins found in silk. They have a certain relationship but have significant differences in structure and function [145]. Sericin is a gel-like protein composed of various members of the sericin protein family, such as sericin 1 and sericin 2 [146]. Its structure is relatively irregular and contains many polar amino acids and hydrophobic amino acids, giving it excellent hydrophilicity and colloidal properties [147]. On the other hand, fibroin provides silk with excellent mechanical properties, tensile strength, and flexibility. Sericin primarily plays a protective and adhesive role.

#### 2.2.2. Synthetic Polymers

(a)Polylactic acid (PLA)

Polylactic acid (PLA) is a renewable, low-cost, manufacturable, and biodegradable synthetic material. PLA is synthesized from L and D lactic acid monomers as an aliphatic hydrophobic polyester. Standard synthesis methods include dehydration condensation, direct condensation, and ring-opening polymerization (ROP) [148]. PLA can be hydrolyzed, enabling its natural degradation within the human body, and its byproduct—lactic acid—is nontoxic [149]. PLA degradation is relatively slow, primarily due to its hydrophobic methyl groups. The mechanical properties of PLA are determined by its degree of crystallinity, which is influenced by the molecular weight and stereochemical properties. Generally, as crystallinity increases, the toughness and stiffness of PLA also increase. Nevertheless, PLA is very brittle and has a poor capacity for plastic deformation, which limits its application in tissue engineering. Additionally, as a hydrophobic material, PLA can cause low cell attachment, potentially leading to new inflammatory responses. To optimize PLA’s applicability in tissue engineering, it can be blended with other polymers such as polyethylene glycol, polyglycolic acid, and polycaprolactone to improve the flexibility, hydrophilicity, and elasticity of the resulting product. Increasing the number of PLA nanofiber layers in scaffolds can also enhance their mechanical properties [150,151].

(b)Polyglycolic acid (PGA)

Polyglycolic acid (PGA) is considered to be the first biodegradable synthetic polymer in the field of biomedical sciences and is utilized in a biodegradable synthetic suture product called “DEXON” [152]. Due to the absence of a methyl group, PGA possesses hydrophilic properties and can degrade through hydrolysis. With low solubility, high crystallinity, and strong mechanical strength, PGA can maintain its shape in organic solvents while demonstrating controllable degradation properties [153]. Although its rapid degradation capability is beneficial for short-term implants, it also limits its application, as degradation within a few weeks can affect the mechanical performance of long-term implants [154,155]. The toxicity of its degradation byproducts is an issue that also needs to be addressed when using PGA, as the glycolic acid produced by hydrolysis can acidify the local environment and lead to inflammation [156]. To optimize PGA, copolymerizing it with polylactic acid (PLA) through a copolymerization reaction to form a polylactic acid–coglycolic acid (PLGA) copolymer is a common approach, enhancing its mechanical properties and regulating the degradation rate.

(c)Polycaprolactone (PCL)

Polycaprolactone (PCL) possesses various characteristics, such as good biocompatibility, biodegradability, and nontoxicity. There are two main methods for the synthesis of PCL: The ring-opening polymerization (ROP) of ε-caprolactone and the polycondensation of 6-hydroxy hexanoic acid [157,158]. Of these, the ROP pathway is the most commonly used method. PCL has a semicrystalline structure and, under physiological conditions, it maintains high mechanical properties (e.g., elasticity and toughness). The degradation mechanism of PCL implants typically involves hydrolysis of ester linkages and digestion by resident micro-organisms, with a general degradation period of about 2–3 years [159,160]. The degradation product of PCL—caproic acid—is nontoxic to local tissues and can be excreted through the renal system or reabsorbed during metabolic processes [161]. Low cell adhesion, slow degradation, and mechanical property mismatches are the main drawbacks of PCL, and blending PCL with other polymers is the primary approach used to overcome these challenges [162].

(d)Poly(lactic-co-glycolic) acid (PLGA)

Poly(lactic-co-glycolic acid) (PLGA) is a copolymer of PGA and PLA. By copolymerizing PGA and PLA, both of their properties can be better customized, resulting in PLGA with controllable degradation time, excellent processability, good biocompatibility, and commercial availability [163]. PLGA degrades into lactic acid and glycolic acid through bulk erosion and hydrolysis of its ester bonds, which are then metabolized by the body through natural metabolic pathways and, ultimately, converted into carbon dioxide and water. By adjusting the ratio of LA to GA in PLGA, materials with the desired degradation rate and mechanical properties can be obtained to meet the requirements of various biomedical applications. When the LA content in PLGA is higher, the degradation rate slows down as the ester bonds of the LA monomer are more stable during hydrolysis, resulting in a slower degradation rate. Conversely, when the GA content in PLGA is higher, the degradation rate accelerates as the ester bonds of the GA monomer are more easily hydrolyzed. When the lactide:glycolide ratio is 1:1, the copolymer has been shown to degrade within 1–2 months, while the degradation time increased to 5–6 months for the 85:15 blend [152]. PLGA nanofibers have been shown to contract in fluids, including simulated body fluid (SBF), Dulbecco’s Modified Eagle Medium (DMEM), and saliva. This volume erosion occurring in PLGA leads to a decline in its mechanical properties as it degrades.

#### 2.2.3. Multipolymer Blends

Natural polymers tend to be biocompatible and biodegradable with natural bioactivity, but their poor mechanical properties and significant batch-to-batch variation in extraction and preparation limit their application in the tissue-engineering field. Synthetic polymers, on the other hand, tend to have good adjustable mechanical properties and good reproducibility, but their cytotoxicity, lack of synchronization of degradation rates with tissue repair, and lack of natural bioactivity make the preparation of nanofiber dressings equally challenging. Electrostatic spinning is the most commonly used technique for preparing nanofibers; however, some natural polymers, such as chitosan, are not spinnable on their own and cannot produce stand-alone electrospun films when used as a raw material. Therefore, to combine their advantages, overcome some of the disadvantages of the single substrates, and enable their preparation by electrostatic spinning, such natural polymers may be blended with synthetic polymers or other natural polymers. In Table 2, we list some composites using a variety of polymers as raw materials.

### 2.3. Drug Loading Method of Nanofiber Scaffold

#### 2.3.1. Physical Adsorption Method

The physical adsorption method is a drug-loading technique that involves adsorbing drugs onto the scaffold surface through physical interactions. This approach utilizes nonchemical bond forces, such as van der Waals forces and electrostatic interactions, between the drug and the carrier to achieve drug adsorption [62]. To load drugs using the physical adsorption method, the drug must first be dissolved in an appropriate solvent to prepare a drug solution. The scaffold is then immersed in the drug solution, allowing the drug to attach to the scaffold surface through physical interactions. The adsorption time should be chosen based on the properties of the drug and scaffold, as well as the desired drug-loading capacity. After adsorption, the drug-loaded scaffold is removed from the drug solution, washed with pure solvent or physiological saline to remove the residual drug solution, and dried. The main advantage of the physical adsorption method is its simplicity, as it does not alter the drug’s chemical structure or activity. As there is no chemical bonding between the drug and the scaffold, the drug’s chemical structure and activity remain unaffected [185]. Therefore, the physical adsorption method offers high versatility. However, due to the physical interaction between the drug and the scaffold, the drug release rate may be relatively fast, which is not conducive to sustained or controlled release. Additionally, the drug-loading capacity provided by the physical adsorption method is limited by the surface area of the scaffold and the concentration of the drug solution, possibly hindering high drug loadings. Moreover, drug-loaded nanofiber scaffolds prepared using the physical adsorption method may be more sensitive to environmental conditions (e.g., temperature, humidity, and pH), potentially resulting in poor adsorption stability. Fatemeh Koohzad et al. combined hyaluronic acid–chitosan–polyvinyl alcohol composite nanofibers with a net positive charge at physiological conditions with temporin-Ra peptide, which has a negative surface charge at physiological pH through physical adsorption. This process enabled the fixation of peptide molecules on the scaffold [81]. Due to the alkaline environment of the wound bed, the release of the peptide from the nanofiber scaffold can be controlled by pH [186].

#### 2.3.2. Chemical Conjugation Method

The chemical conjugation method is a drug delivery approach that involves connecting a drug to a molecular scaffold structure through the use of a chemical reaction. This method forms stable chemical bonds between the drug and the scaffold (e.g., covalent bonds, ester bonds, and amide bonds). When using the chemical conjugation method for drug delivery, appropriate functional groups (e.g., carboxyl, amino, and hydroxyl) must be selected for the chemical reaction [187]. First, the drug or scaffold itself must have a reactive functional group or be modified with a functional group to achieve chemical bonding of the scaffold to the drug [188]. Then, under suitable conditions (e.g., temperature, time, solvent, and catalyst), the chemical reaction is carried out, binding the drug to the scaffold. Finally, unreacted drugs and impurities are removed through purification and washing to obtain the drug-conjugated scaffold.

Compared to physical adsorption methods, chemical conjugation can achieve more stable drug–scaffold binding, which is beneficial for slow, sustained, and controlled drug release. Changing the reaction conditions and functional group types can regulate the degree of drug binding and release rate. Moreover, this method allows for the codelivery of multiple drugs to achieve combined therapeutic effects [189]. However, the chemical conjugation method has some limitations, such as potential alteration of the drug’s chemical structure and activity, more stringent reaction conditions, a relatively complex preparation process, and the possible requirement of toxic or highly active reagents [190].

#### 2.3.3. Coating Method

Coating is a drug-loading method that involves covering the surface of a scaffold with a drug or drug–carrier composite. To carry out the coating method, the drug must be dissolved in an appropriate solvent to prepare a drug solution [191]. An appropriate concentration for the drug solution can be selected based on the properties of the drug and scaffold. The scaffold can be immersed in the drug solution or coated with the drug solution on its surface using methods such as spraying, roll-coating, or brush-coating, ensuring a uniform distribution of the drug on the scaffold’s surface [192]. The drug solution must be thoroughly dried on the scaffold surface during the drug fixation process in order to form a uniform drug coating layer. Various methods, such as room temperature, hot air, or vacuum drying, can be used. Finally, the drug-coated scaffold is removed from the drug solution and rinsed with a pure solvent or saline to remove any residual drug solution [193]. The advantages of the coating method include simple operation, ease of achieving a uniform drug distribution on the scaffold surface, the ability to load multiple drugs for combined therapeutic effects, and the ability to regulate the drug loading and release rate by adjusting factors such as the drug solution concentration and coating method [194]. However, due to the weak binding between the drug and the scaffold in the coating method, the drug release rate may be too fast, which is not conducive to achieving sustained and controlled release [195]. It is worth mentioning that the physisorption method is more suitable for small molecules, where the drug is attached to the scaffold by van der Waals forces, whereas the encapsulation method is more suitable for large molecules (e.g., proteins), which mainly involves trapping the drug molecules within the pores of the material [196,197,198].

#### 2.3.4. Coblending Electrospinning Method

Blended electrospinning allows for the fabrication of drug-loaded nanofiber scaffolds by combining drugs with scaffold materials and then utilizing electrospinning technology [63]. This approach allows for excellent biocompatibility and mechanical properties, making them suitable for various biomedical applications [199]. The specific steps of blended electrospinning include dissolving the drug in an appropriate solvent to prepare a drug solution, dissolving the scaffold material (e.g., polylactic acid, polycaprolactone, or chitosan) in a corresponding solvent to prepare a scaffold-material solution, mixing the drug solution with the scaffold-material solution at a specific ratio to form a drug-loaded blending solution, and placing the drug-loaded blending solution into an electrospinning device [66]. Drugs and scaffold materials are spun together during electrospinning, forming nanofiber scaffolds with a uniform drug distribution [200]. The performance of drug-loaded nanofibrous scaffolds can be tuned by adjusting parameters such as the voltage and nozzle-to-collector distance. Finally, the resulting drug-loaded nanofiber scaffolds are collected and dried to remove the residual solvent.

The advantages of blended electrospinning include uniform drug distribution within the nanofiber scaffold, facilitating slow and sustained release, good biocompatibility, and improved mechanical properties. However, blended electrospinning has notable limitations, such as compatibility issues between drugs and scaffold materials, drug stability during the spinning process, difficulties in controlling drug release rates, and challenges in scaling up production. Drug–scaffold material compatibility can impact the spinning process and drug release behavior. Poor compatibility may result in solution instability during spinning, affecting nanofiber formation and quality, as well as leading to uneven drug distribution within the scaffold, thus affecting the drug release behavior. Some drugs may encounter stability issues such as decomposition or degradation during spinning. This is particularly true for thermally sensitive or photosensitive drugs, as solvent evaporation during electrospinning may cause temperature increases or exposure to light, thus affecting the stability of the drug. Although blended electrospinning can achieve slow, sustained, and controlled release, precisely controlling drug release rates and durations is still challenging. This requires adjusting various factors, such as the drug–scaffold material blending ratio and spinning parameters [201]. Furthermore, while electrospinning technology is easily operable at the laboratory scale, its application at industrial-production scales remains challenging. Addressing equipment scaling and increased spinning speeds is necessary to meet large-scale production demands [202].

## 3. Application of Nanofibrous Scaffolds in Wound Healing

### 3.1. Promotion of Hemostasis

The hemostatic function of drug-loaded nanofiber scaffolds is typically achieved by incorporating bioactive molecules, such as thrombin, clotting factors, and hemostatic peptides, into the nanofiber scaffold [203]. Certain natural polymers, such as gelatin and chitosan, also possess good hemostatic properties and can be prepared as nanofiber scaffolds in the form of hemostatic sponges, which can rapidly achieve hemostasis and localized drug release when loaded with drugs [204,205]. Thrombin is an enzyme that acts directly on fibrinogen, converting it into fibrin and accelerating the blood-clotting process. Giriraj Pandey et al. designed a polyvinyl alcohol/gelatin/poly(lactic-co-glycolic acid) nanofiber scaffold rich in thrombin (TMB) and vancomycin (VCM) to control excessive bleeding, inhibit bacterial growth, and promote wound healing. In a rat model, the thrombin-loaded nanofiber scaffolds exhibited shorter bleeding time, rapid clotting, and excellent wound closure [98]. Hemostatic peptides are a class of bioactive peptides that promote platelet aggregation, fibrin formation, and vasoconstriction. Marta A. Teixeira et al. loaded the hemostatic peptide Tiger 17 onto a polyvinyl alcohol (PVA) nanofiber mat reinforced with cellulose nanocrystals (CNC) produced by electrospinning, which showed an excellent ability in terms of accelerating clotting [206]. Some ions can activate multiple enzymes required for blood coagulation and play an essential role in the blood clotting process, such as aluminum and calcium ions. Xinrong Yu et al. designed a gelatin–calcium chloride electrospun nanofiber for rapid hemostasis. The results of dynamic whole-blood clotting tests, hemolysis tests, platelet adhesion tests, and clotting time tests demonstrated that the calcium-containing gelatin nanofibers had a shorter clotting time and lower hemolysis rate when compared to pure gelatin sponges [207]. Sara Nasser et al. designed a PLLA nanofiber mat loaded with aluminum chloride, which presented an optimal hemostatic performance at 30% w/w compared to the gauze-bandage group, reducing the blood coagulation time by 80% and increasing the blood absorption capacity by 178% [66].

### 3.2. Reduction in Chronic Inflammation

Chronic inflammation is primarily caused by proinflammatory chemokines, high levels of reactive oxygen species (ROS), and bacterial infections, leading to delayed wound healing. Modulating the wound inflammatory microenvironment can be achieved by eliminating ROS, downregulating the expression levels of inflammatory factors, and regulating the phenotype and number of immune cells. Some herbal extracts with immunomodulatory and anti-inflammatory properties can accelerate wound healing [10], such as *Aloe vera* [208], propolis [68], calendula [209], astragalus [210], cinnamon [211], dihydromyricetin [212], and polyphenols [213]. Zijian Wang et al. developed a multilayer nanofiber (referred to as DQHP-n, n = 0, 2, 6, and 10) incorporating dihydromyricetin (DHM) using the layer-by-layer (LBL) self-assembly technique. DQHP-6 presented the best performance, promoting cell migration by eliminating inflammation, clearing reactive oxygen species, promoting hemostasis and sterilization, and reconstructing the harsh wound microenvironment [212]. In the absence of drug loading, optimizing the raw material composition and spatial structure of nanofibers can also promote the healing of chronic inflammatory wounds. Qingchang Chen et al. developed a three-layer nanofiber membrane (nBG-TFM) incorporating chitosan, PVA (polyvinyl alcohol), and bioactive glass through sequential electrospinning technology. Optimization of the spatial design structure enhanced the functionality of each component, allowing the membrane with a three-layer spatial structure to provide a more suitable microenvironment for diabetic wounds in mice. Experiments showed that nBG-TFM up-regulated factors including TGF-β and VEGF, while the inflammatory cytokines IL-1β and TNF-α were downregulated [214]. In another experiment, Hao Yu et al. designed a porous poly(L-lactic acid) (PLA) nanofiber membrane and sequentially modified its surface with sulfated chitosan (SCS) and polydopamine–gentamicin (PDA-GS). PDA was applied to the PLA nanofiber membrane to impart an anti-inflammatory effect. In vitro cell testing results demonstrated that the PLA/SCS/PDA-GS membrane could immunomodulate macrophages toward M2 phenotype development. Moreover, the PLA nanofiber membrane containing SCS also exhibited an enhanced potential for promoting macrophage-to-fibroblast transdifferentiation [215].

Interestingly, the alignment of fibers in the dressing also has an impact on the cellular response, which is correlated with microenvironmental cues in the healing process. Luyao Sun et al. fabricated nanofibrous scaffolds with crossed fiber organization, and the crossed nanofibrous scaffolds induced different cell morphology, cell migration, and wound-healing-related effects in fibroblasts, compared to aligned or random nanofibrous scaffolds, according to gene expression profiles. In a diabetic rat system, crossed nanofiber scaffold-treated wounds exhibited optimal healing, as evidenced by the regression of inflammation, accelerated migration of fibroblasts and keratin-forming cells, and promotion of angiogenesis [216].

### 3.3. Enhancement of Cellular Proliferation, Migration, and Secretion

During the wound-healing process, normal fibroblasts at the wound edges proliferate and migrate toward the wound site while simultaneously secreting fibrous tissue and matrix components, such as collagen, elastin fibers, and polysaccharides. In particular, collagen provides structural support for newly formed cells, facilitates cell-to-cell interactions and tissue repair, and serves as a substrate for cell migration, allowing cells to move toward the center of the wound during the healing process. In addition, collagen plays a positive role in promoting angiogenesis and regulating inflammatory responses. Therefore, promoting the proliferation, migration, and secretion of wound tissue cells has significant implications in the field of tissue engineering.

Growth factors are a class of proteins that can stimulate cell proliferation, differentiation, and migration. Growth factors applied in the wound-healing process include epidermal growth factor (EGF), fibroblast growth factor (FGF), vascular endothelial growth factor (VEGF), and transforming growth factor-β (TGF-β). These growth factors can promote fibroblast proliferation and differentiation, as well as increasing collagen production. Chen Wang et al. constructed a quaternary ammonium salt-grafted sulfated chitosan scaffold modified with polydopamine and loaded with bioactive EGF and bFGF. In a full-thickness skin-defect mouse model, sustained release of dual growth factors (EGF/bFGF) was achieved, enhancing collagen deposition and promoting angiogenesis and hair follicle regeneration at the injury site [217].

Some vitamins are also crucial for wound healing, such as vitamin C [218], vitamin A [219], vitamin E [218], vitamin B12 [220], and vitamin B2 [221]. Vitamin A promotes the production of type I collagen and fibronectin in the extracellular matrix, increases the proliferation of fibroblasts and keratinocytes, and reduces the levels of matrix metalloproteinases [222]. Vitamin C aids in collagen synthesis and has a synergistic effect with vitamin E in promoting collagen synthesis and wound closure [218]. Sayeed Farzanfar et al. designed a poly(caprolactone)/gelatin nanofiber scaffold loaded with vitamin B12 [220]. Cell experiments showed a significant increase in L929 cell proliferation at 1 and 3 days postseeding. After 14 days, the dressings containing vitamin B12 significantly enhanced wound closure (92.27 ± 6.84% vs. 64.62 ± 2.96%), compared to the group without vitamin B12, and histopathological examination revealed a significant increase in epithelial thickness in the B12-loaded group.

Curcumin—a bioactive substance that has recently gained widespread attention—has been shown to promote fibroblast proliferation and collagen deposition. Jiya Jose et al. used cellulose nanofibers loaded with curcumin, demonstrating a promotive effect on fibroblast migration in a chronic wound model [223].

### 3.4. Neovascularization Support

Angiogenesis is essential for wound healing and regeneration. Adequate angiogenesis provides nutrients, oxygen, and bioactive substances to the wound while assisting in the removal of metabolic waste [224]. Drug-loaded nanofiber scaffolds can carry a variety of angiogenic-promoting drugs, such as growth factors, cytokines, and collagen proteins. By loading these angiogenesis-promoting agents, drug-loaded nanofiber scaffolds can effectively improve blood supply to the wound site, accelerating the wound-healing process.

Vascular endothelial growth factor (VEGF) is the most common growth factor incorporated into nanofiber scaffolds for the promotion of angiogenesis [185]. Other growth factors, such as fibroblast growth factor (FGF) [195], platelet-derived growth factor (PDGF) [225], and transforming growth factor-β (TGF-β) [226], can also synergistically promote angiogenesis. Some cytokines, such as monocyte chemoattractant protein-1 (MCP-1) and interleukin-8 (IL-8) [227,228], can induce endothelial cell migration and proliferation, thereby promoting angiogenesis. Collagen proteins possess excellent biocompatibility and biodegradability and can serve as a matrix for angiogenesis [229]. As such, drug-loaded nanofiber scaffolds containing collagen proteins can provide an ideal growth environment for newly formed blood vessels [230]. Anqi Zhan et al. designed a multifunctional CTS@PLCL/DWJM@Cu core–shell nanofiber loaded with three bioactive agents, including chitosan (CTS), copper (Cu), and decellularized Wharton’s jelly matrix (DWJM), which exhibited continuous antibacterial, angiogenic, and collagen deposition activities. In vitro analysis showed that, after the initial release of the CTS shell, the sustained release of the copper core enhanced the expression of angiogenesis-related genes, promoting cell migration and the formation of new blood vessels [231]. To promote angiogenesis at the wound site, Mohamadreza Tavakoli et al. prepared core–shell nanofibers with polyvinyl alcohol (PVA) cores and gelatin (Gel) shells loaded with advanced platelet-rich fibrin (A-PRF) using the coaxial electrospinning method. A chorioallantoic membrane (CAM) assay revealed that the PVA/(Gel/A-PRF) samples demonstrated the highest angiogenic potential among all sample groups [200].

### 3.5. Combating Bacterial Infections

For the treatment of infected wounds, antibiotics, metal and metal oxides, antimicrobial peptides, lysozymes, and antimicrobial polymers can be loaded into nanofiber scaffolds to enhance their antibacterial performance and combat bacterial and fungal infections [202,232,233]. Antibiotics have strong bactericidal properties and rapid efficacy. Local administration at the site of infection can achieve good therapeutic effects. Combining antibiotics with nanofiber scaffolds can achieve localized drug delivery and slow release. Nur Adila Mohd Razali et al. fabricated a sandwich-like composite scaffold for wound healing composed of a hydrogel layer and two aligned nanofiber layers. Gentamicin was loaded into the middle hydrogel layer, exerting its antibacterial activity through natural release, while the outer nanofiber layers of the hydrogel act as diffusion barriers, reducing the initial burst release of gentamicin (15–30%) [234]. Hao Yu et al. modified poly(L-lactic acid) (PLA) porous nanofiber membranes by incorporating polydopamine–gentamicin (PDA-GS) and sulfated chitosan (SCS), promoting the controlled release of GS through dopamine self-polymerization to prevent early wound infection [215].

Compared with antibiotics, metal nanoparticles, metal ions, and metal oxides have advantages in preventing bacterial resistance. Nano-Ag [235], nano-Cu [236], nano-Au [192], Ag^+^ [237], Zn^+^ [238], Cu^2+^ [239], Mg^2+^ [237], CuO [240], and Ag_2_O [241] may all be loaded into nanofiber scaffolds to exert antibacterial properties.

Antimicrobial peptides (AMPs) are small peptides with broad-spectrum antimicrobial activity in living organisms, capable of regulating the host immune response while controlling microbial reproduction and colonization. For example, Temporin-Ra peptides, which are mainly found in the skin secretions of amphibians (e.g., frogs and toads), play a crucial role in combating pathogenic microbes. They usually act as the first line of defense in living organisms, protecting them from pathogen invasion. Temporin-Ra possesses various biological activities, such as antibacterial, anti-inflammatory, and wound-healing promotion. Due to its low cytotoxicity and high biocompatibility, the Temporin-Ra peptide has extensive potential in drug development and biomedical applications. Fatemeh Koohzad et al. designed an antimicrobial peptide drug-loading platform that utilizes surface adsorption principles and used anionic hyaluronic acid–chitosan–polyethylene glycol composite nanofibers to adsorb Temporin-Ra peptide molecules with a net positive charge under physiological conditions, thereby immobilizing the peptide molecules on the synthesized scaffold. In the alkaline environment of the wound bed, the antimicrobial peptide is released from the nanofiber scaffold in a pH-responsive manner [81].

Lysozyme is a naturally occurring enzyme that can break down the polysaccharide structure of bacterial cell walls, thus exerting an antibacterial effect. Lysozyme is distributed in animals, plants, and micro-organisms, and, in the human body, it is present in saliva, tears, nasal secretions, breast milk, and other body fluids. Jun Wu et al. incorporated lysozyme (LY) into silk fibroin/polycaprolactone nanofiber mats by electrospinning, then deposited chitosan and polydopamine (PDA) onto the surface through layer-by-layer (LBL) self-assembly. As the nanofibers degrade at the wound site, LY is gradually released. CS rapidly dissociates from the nanofiber mats, synergistically providing effective inhibition of *Staphylococcus aureus* (*S. aureus*) and *Escherichia coli* (*E. coli*) [242].

Antimicrobial polymers mainly refer to high-molecular-weight polymers with inherent antibacterial properties, typically featuring long-lasting and low-toxicity antibacterial performance. These polymers, such as polyquaternary ammonium salts and polyamine antimicrobial polymers, achieve antibacterial functions through chemical structure regulation or surface modification. Hiroyuki Kono et al. designed quaternary ammonium salt-modified bacterial cellulose nanofibers, with nanofibrillated bacterial cellulose (NFBC) modified through an alkaline aqueous solution of 2,3-epoxypropyltrimethylammonium chloride (EPTMAC) to impart antibacterial properties [243].

### 3.6. Inhibition of Scar Formation

Scars are fibrous tissues formed during the wound-healing process, primarily produced by fibroblasts. By altering the activity and proliferation of fibroblasts, scar formation can be suppressed; for example, with common antifibrotic drugs such as mitomycin C and 5-fluorouracil (5-FU) [244]. Hongmei Zhang et al. designed a dendritic mesoporous bioactive glass nanoparticle (dMBG) loaded with 5-fluorouracil (5-Fu), coaxially electrospun with polyethylene oxide (PEO)-poly(ether-ester-urethane) urea (PEEUU) shell–composite nanofibers ((F@B)/P)@PU), which exhibited antiproliferative activity by inhibiting the growth of HeLa cells. In a dorsal rat experiment, the fiber scaffold loaded with 5-FU was shown to inhibit hypertrophic scar formation [245].

Curcumin has also been combined with nanofiber scaffolds for scar treatment. Vivek Kumar Pandey et al. designed a polyvinylpyrrolidone (PVP)-cerium(III) nitrate hexahydrate (Ce(NO3)3.6H2O) and curcumin composite nanofiber scaffold synthesized using electrospinning technology. When applied to a full-thickness defect wound model in rats, complete healing and re-epithelialization were achieved within 20 days without scarring. The antiscarring characteristics of the PVP–Ce–Cur NFs were also demonstrated by measuring the regulatory levels of catalase and hydroxyproline superoxide dismutase (SOD) in granulation tissue [246].

Astragalus species possess antibacterial and anti-inflammatory properties that can accelerate wound healing. To alleviate scars, Danping Zhang et al. used a silk fibroin/gelatin (SF/GT) electrospun nanofiber dressing loaded with astragaloside IV (AS) for acute wounds. Compared to the control group, the SF/GT nanofiber dressing loaded with AS inhibited the expression level of α-SMA, thus reducing the scarring effect [201].

Lovastatin has also been found to help reduce scars. Zuhan Chen et al. prepared poly(caprolactone)/silk fibroin electrospun nanofiber membranes loaded with or without lovastatin, then evaluated their inhibitory effect on scars under specific tension directions. The results confirmed that the nanofiber membrane loaded with lovastatin placed vertically with respect to the wound tension direction reduced the scar area by 66.9%, most effectively reducing scar formation [181].

## 4. Multiresponse Intelligent Drug Delivery Systems Based on Nanofibrous Scaffolds

Wound healing is a spatiotemporally ordered physiological process and developing novel smart dressings with controlled drug release in both time and space is a higher pursuit in the field of tissue engineering. Bioresponsive drug delivery nanofiber scaffolds are a type of drug delivery system that can release therapeutics in response to specific biological signals or environmental changes. The primary principle of these scaffolds is to provide highly targeted and localized control in therapeutic drug release, thus enhancing the efficacy and safety of treatments. Bioreactivity largely depends on the properties of the polymers used to construct the scaffolds, including their pH responsiveness, temperature responsiveness, light responsiveness, electric-field responsiveness, and magnetic-field responsiveness [247,248].

### 4.1. pH-Responsive Nanofiber Drug Delivery Systems

pH-responsive polymers often have easily ionizable functional groups, such as carboxyl and amine groups. When the environmental pH changes, these functional groups undergo ionization or protonation, thus causing the polymer chains to swell or contract and altering the scaffold’s pore structure [249]. Polyethyleneimine (PEI) contains amino groups which undergo protonation under acidic conditions, leading to swelling, and de-protonation under alkaline conditions, resulting in contraction [250]. Some polymers may undergo hydrophobic–hydrophilic transitions at different pH values. This transition can change the interactions between the scaffold and the drug, thus affecting drug release. N-isopropylacrylamide (NIPAM) is a neutral polymer, the water solubility of which is affected by the pH of the solution, allowing it to precipitate from the aqueous solution at specific temperatures and resulting in a hydrophilic–hydrophobic transition [250]. This structural change enables the diffusion and release of drugs within the scaffold to be regulated by pH. Changes in pH may also alter the solubility of polymer scaffolds, leading to their partial or complete dissolution and consequent drug release. For example, certain polyester materials may be more prone to degradation under acidic conditions, thereby accelerating drug release. Alternatively, polymer scaffolds may contain acid- and/or base-sensitive chemical bonds. Changes in pH may cause these bonds to break, leading to structural changes in the scaffold and drug release. For example, ester and amide bonds are prone to hydrolysis under acidic or alkaline conditions, leading to drug release [251]. These changing mechanisms allow pH-responsive nanofiber scaffolds to achieve targeted drug release; for example, in tumor tissue or at inflammation sites [252]. This helps to improve their therapeutic efficacy and reduce side effects [253]. Fatemeh Koohzad et al. designed a nanofibrous scaffold that uses electrostatic action to adsorb drugs, enabling the slow release of drugs at alkaline wound sites (Figure 2) [80].

Ismail Altinbasak et al. reported a pH-responsive nanofiber scaffold synthesized by electrospinning using a polymer composed of a hydrolysable, acid-sensitive, and side-chain trimethoxybenzaldehyde-protected acrylate main chain. At physiologically relevant pH = 6.5 in the extracellular tumor environment, the scaffold transitions from a hydrophobic state to a hydrophilic state, leading to hydrolysis of the side chains and an approximately three-fold increase in fiber diameter, thereby releasing the drug encapsulated within the nanofiber scaffold [254].

### 4.2. Temperature-Responsive Nanofiber Drug Delivery Systems

Temperature-responsive nanofiber scaffold mechanisms mainly involve phase transitions, expansion, degradation, or weakening of the connections between scaffold materials and drugs as the temperature increases. Some temperature-sensitive polymers undergo phase transitions near their lower critical solution temperature (LCST) or upper critical solution temperature (UCST); for example, poly(N-isopropylacrylamide) (PNIPAM) is a temperature-sensitive polymer with an LCST. When the temperature rises to approximately 32 °C, PNIPAM changes from hydrophilic to hydrophobic, causing the scaffold to shrink and release the drug. Simultaneously, an increase in temperature may cause thermal expansion of the polymer scaffold [255]. This expansion can alter the pore structure of the scaffold, increasing drug diffusion pathways and rates and leading to rapid drug release. In some cases, a temperature increase may lead to thermal degradation of the polymer scaffold. This degradation can damage the scaffold structure and release the drug. For example, biodegradable polymers, such as polylactic acid (PLA) and polycaprolactone (PCL), are more likely to degrade at higher temperatures. Additionally, an increase in temperature may cause changes in the interactions between polymers and drugs within the scaffold. For example, hydrogen bonds or van der Waals forces between the polymer and drug may weaken at higher temperatures, leading to the release of drugs. Temperature-responsive nanofiber scaffolds can achieve controlled drug release at specific temperatures, helping to improve treatment outcomes, reduce side effects, and allow for their application in thermosensitive disease treatments, such as inflammation and infection [252].

Xiaocheng Wang et al. utilized the photothermal conversion ability of MXene nanosheets and the temperature-responsive contraction/expansion behavior of poly(NIPAM) hydrogels to realize the controllable delivery of vascular endothelial growth factor (VEGF), in order to promote endothelial Cell proliferation, migration, and proangiogenic effects (Figure 3) [256]. Yuting Liang et al. designed a biocompatible and intelligent dual-responsive cellulose nanofiber by grafting the temperature and pH dual-responsive polymer polyethyleneimine-N-isopropyl acrylamide (PEI-NIPAM) onto cellulose nanofibers (CNF-COOH). When loaded with actinomycin, the nanofiber demonstrated an accumulated release rate of up to 59.45% at 37 °C [250].

### 4.3. Photoresponsive Nanofiber Drug Delivery Systems

The mechanisms of photoresponsive scaffolds mainly involve the cleavage of photosensitive chemical bonds, photo-induced isomerization, charge transfer, and photothermal effects [257,258]. Polymer scaffolds containing photosensitive chemical bonds (e.g., photocleavable ester bonds or photocleavable amide bonds) undergo chemical bond breaking under light irradiation, leading to the destruction of the scaffold structure and the consequent release of the drug (Figure 4) [257]. Some polymers or photosensitizers undergo isomerization under light exposure, such as cis–trans isomerization. This isomerization may cause changes in scaffold structure and properties, thereby affecting drug release [259]. Charge transfer in some photoresponsive polymers under light exposure may change the interaction between the polymer and the drug, leading to the release of the drug. At the same time, light irradiation may cause local heating of the polymer scaffold, leading to thermal expansion, thermal degradation, or changes in thermally induced interactions, thereby also resulting in drug release [252,260]. For example, nanofiber scaffolds containing gold nanoparticles undergo surface plasmon resonance under light exposure, producing a local thermal effect that releases the drug [261].

Lin Jin et al. prepared MXene nanoribbon fibers (T-RMFs) with vitamin E-possessing controllable temperature-responsive drug-release capabilities. The distribution of MXene nanoplates with polyvinylpyrrolidone and polyacrylonitrile composite nanoribbons and thermosensitive PAAV coating endows the T-RMFs with excellent photothermal properties. The temperature can be controlled through near-infrared (NIR) light, by which the temperature-responsive polymer coating relaxes the interface, dissolving vitamin E and promoting its release [262].

### 4.4. Electroresponsive Nanofiber Drug Delivery Systems

Electrically responsive nanofiber scaffolds mainly involve electrostriction, electric charge-controlled drug desorption, electro-osmosis, and electrolysis processes. Some polymers undergo shape changes under the action of an electric field (i.e., electrostriction). This deformation alters the scaffold’s pore structure, affecting the diffusion rate and release of the drug within the scaffold. The electric field may cause changes in the charge distribution between the polymer scaffold and the drug, leading to adsorption or desorption of the drug on the scaffold, thereby achieving drug release [263]. The electro-osmotic effect refers to the movement of drug molecules or ions within the nanofiber scaffold due to their charge under the action of an electric field, causing the drug to diffuse and release in the direction of the electric field [264]. In addition, electrolytic action can also promote electric-field-sensitive drug release. The polymer scaffold undergoes hydrolysis reactions or chemical bond breaking in the electric field, disrupting the scaffold structure, and enabling drug release. Alexa-Maria Croitoru et al. designed a poly(lactic acid) (PLA)/graphene oxide (GO) microscaffold loaded with quercetin (Q), which was synthesized by electrospinning. When an appropriate electric field was applied, the Q release rate became 8640 times faster (Figure 5) [263].

### 4.5. Magnetic-Responsive Nanofiber Drug Delivery Systems

Magnetic-responsive polymers are typically copolymerized or physically mixed with magnetic nanoparticles. Under the influence of a magnetic field, magnetic-responsive scaffolds exhibit magnetostrictive, magnetothermal, and magnetomotive effects, and/or magnetic-field-induced chemical bond breaking [265,266,267,268]. These changes result in alterations to the structure, chemical properties, or physical properties of the polymer chains, thus achieving magnetically controlled drug release. Die Dong et al. designed a multiresponsive cellulose nanofiber drug delivery system loaded with Fe_3_O_4_ nanoparticles as a magnetic switch, allowing for the controlled release of indocyanine green and cytochalasin under the influence of an external magnetic field [252]. Yonggang Zhang et al. designed a magnetically reactive nanofiber ceramic scaffold for on-demand motility and drug delivery, where the hydroxyapatite (HA)/Fe_3_O_4_ sandwich scaffold was shown to be capable of releasing drugs on demand under the action of an alternating magnetic field (Figure 6) [268].

## 5. Conclusions and Future Perspectives

Skin-tissue trauma is a common surgical issue, often requiring multiple treatments and placing significant burdens on the patient. Traditional therapies have limitations related to efficacy, drug concentration at the wound site, and potential side effects. To address these concerns, multifunctional bioactive wound dressings—particularly those utilizing nanofibers—have become a research hotspot due to their excellent properties for use as skin-tissue engineering scaffolds and drug delivery systems.

Nanofiber preparation techniques include electrospinning, sol–gel, molecular self-assembly, and thermally induced phase separation techniques. Electrospinning is the most common and widely used method for manufacturing nanofiber scaffolds. Materials for nanofiber preparation include natural and synthetic polymers. Natural polymers—mainly composed of macromolecular polysaccharides and protein molecules—have good biocompatibility, degradability, and bioactivity but lack mechanical performance. On the other hand, synthetic polymers are relatively inexpensive, easy to prepare, stable, and have good mechanical properties, but tend to have poor bioactivity and may produce toxic byproducts during degradation. Combining synthetic and natural polymers has become a common approach for the construction of scaffolds with optimized bioactivity and mechanical performance.

Nanofiber scaffolds have a similar spatial structure to the extracellular matrix (ECM), possessing properties such as high water absorption, high interconnected porosity, breathability, and moisture permeability, thus creating an ideal environment for hemostasis, preventing exogenous infections, and promoting cell migration and proliferation, cell respiration, and exudate absorption. The unique physicochemical properties of nanofibers make them ideal drug carriers. Drugs can be loaded onto nanofiber scaffolds through various means, including physical adsorption, chemical bonding, coating, and coblending electrospinning. By delivering growth factors, cytokines, anti-inflammatory and antibacterial drugs, herbal extracts, and other bioactive substances, drug-loaded nanofiber scaffolds can promote hemostasis; reduce chronic inflammation; promote cell proliferation, migration, and secretion; help to form new blood vessels; inhibit scar formation; and fight against bacterial infections, thus promoting wound healing. In addition to drug loading, the introduction of physicochemical gradients in engineered constructs has also received significant attention. Multigradient biomaterials for skin regeneration can be designed by biomimicking the physicochemical properties of different skin layers to enhance wound repair mechanisms [269,270].

Stimulus-responsive nanofiber drug delivery systems provide new ideas for spatiotemporal drug release control. When exposed to stimuli, nanofiber scaffolds can achieve controlled drug release through shape changes, phase transitions, degradation, isomerization, charge transfer, and other changes. However, stimulus-responsive dressings are currently more widely used in hydrogel-based and nanoparticle-loaded drug systems and their application regarding fiber scaffolds has not yet been fully explored. Injectable hydrogels for local drug delivery are considered necessary to provide the ability to sense environmental changes occurring in the disease state and should be able to release the optimal amount of a drug to the target area within an appropriate time period. Stimulus responsiveness is, therefore, a key focus for injectable hydrogels and the existing research in this area provides ample reference and inspiration for the exploration of stimulus responsiveness in nanofiber scaffolds [271]. Nanofiber scaffolds can also be combined with other local drug delivery strategies to construct more complex drug delivery systems, such as combining stem-cell engineering technology to culture mesenchymal stem cells on nanofiber scaffolds and then implant them into the body, connecting nanofiber scaffolds with hydrogel systems, or incorporating vesicle delivery systems into nanofiber scaffolds to further achieve targeted drug delivery. It is worth mentioning that the combination of nanofibers and hydrogels has a high potential and takes various forms. In addition to composing separate nanofiber dressing layers with hydrogel layers to form a multilayer structure, nanofibers can also be blended into the hydrogel to serve as one of the components of the hydrogel [272]. For example, Hongyun Xuan et al. blended carboxymethyl-functionalized polymethyl methacrylate (PMAA) short nanofibers with carboxymethyl chitosan, which was blended with aldehyde-functionalized sodium alginate to prepare an injectable self-healing polysaccharide hydrogel [273], and Zhaozhao Ding et al. designed a silk nanofiber hydrogel containing DFO for use in the treatment of diabetic wounds, achieving sustained release of DFO for over 40 days [274]; Vinay Kumar et al. have designed a three-layer nanofibrous scaffold with antibacterial and antioxidant properties, consisting of a top layer of hydrophobic serine protein mixed with polyvinyl alcohol (PVA), an intermediate layer of serine protein loaded with silver sulfadiazine, and a bottom layer of polycaprolactone (PCL) mixed with serine protein [275]. Nanoparticles are also a promising research direction. Metal and metal oxide nanoparticles such as Ag, Cu, Fe, Au, TiO_2_, Fe_3_O_4_, and ZnO have been investigated for the treatment of wound infections [276] and magnetic metal nanoparticles such as Fe_3_O_4_ have been used in the construction of magnetically responsive drug delivery systems [268]. Many bioactive substances, such as curcumin [277], propolis [278], and keratin [279], can be prepared as part of nanoparticles to promote wound healing. In addition to the diversity of types and functions of nanoparticles, the choice of the matrix for nanoparticle loading and the interactions between the nanoparticles and the matrix are also interesting research lines, which could lead to more controlled and efficient drug release.

Nanofiber dressings have been used in clinical applications, such as the CE marked FibDex^®^ (NB0044) from the UPM-Kymmene Corporation [280]. This is a nanofiber dressing prepared from cellulose that improves wound-healing efficiency by maintaining an optimal moisture balance for wound healing, which detaches itself from the wound after epithelialization. There are no established products for drug-carrying nanofiber dressings at the moment, making it a promising direction for development. The use of a fixed nanofiber dressing as a substrate, which is later loaded with the desired drug—for example, by means of a coating method—is a promising idea for clinical applications, which could allow different dressings to be customized to meet the specific needs of the wound site.

## Figures and Tables

**Figure 1 pharmaceutics-15-01829-f001:**
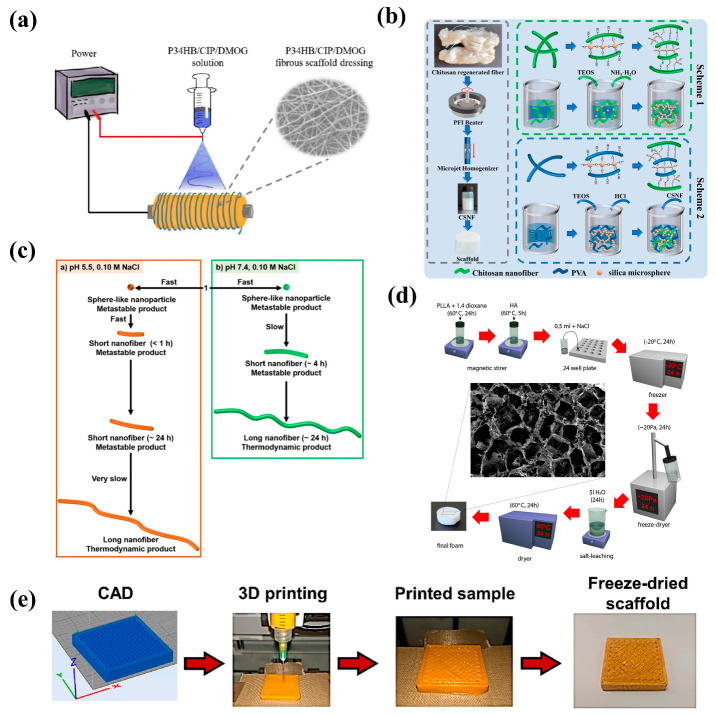
Schematic illustrations of various approaches for the fabrication of nanofibrous scaffolds: (**a**) Electrospinning. Reprinted with permission from Ref. [23] (copyright 2023, Li); (**b**) Sol–Gel method. Reprinted with permission from Ref. [24] (copyright 2022, Ma); (**c**) Molecular self-assembly technology. Reprinted with permission from Ref. [25] (copyright 2023, Hisamatsu); (**d**) Thermally Induced Phase Separation (TIPS). Reprinted with permission from Ref. [26] (copyright 2019, Szustakiewicz); and (**e**) 3D-Printing technology. Reprinted with permission from Ref. [27] (copyright 2021, Olmos-Juste).

**Figure 2 pharmaceutics-15-01829-f002:**
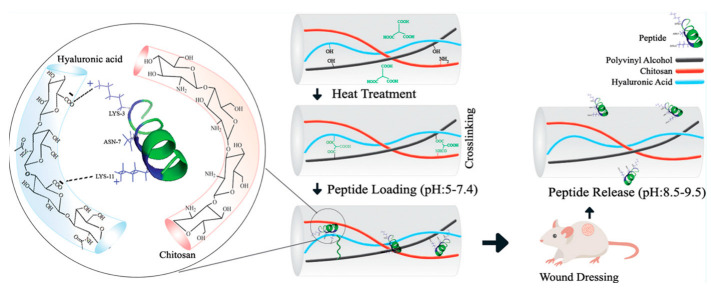
Schematic diagram of cross–linked electrospun pH–sensitive nanofibers adsorbed with Temporin–Ra peptide. A peptide molecule with a net positive charge under physiological conditions is adsorbed and immobilized on a composite nanofiber formed by green cross–linking of hyaluronic acid–chitosan–polyvinyl alcohol with citric acid which is negatively charged at physiological pH. The peptide can be released from the nanofiber scaffold in a wound bed under an alkaline environment. Reprinted with permission from Ref. [80] (copyright 2023, Koohzad).

**Figure 3 pharmaceutics-15-01829-f003:**
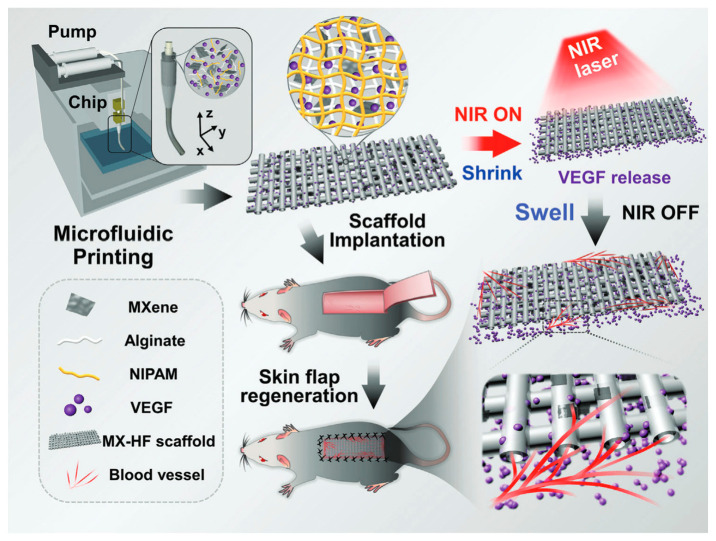
Schematic illustration of the dynamically responsive MXene-incorporated hollow fibrous (MX-HF) scaffolds from microfluidic 3D printing for skin-flap regeneration. The MX-HF scaffold fabricated by a coaxial capillary microfluidic strategy exhibits photothermal-responsive contraction/expansion behavior under near-infrared irradiation, allowing for controllable VEGF release while promoting cell or tissue infiltration into the scaffold channels. Reprinted with permission from Ref. [256] (copyright 2022, Wang).

**Figure 4 pharmaceutics-15-01829-f004:**
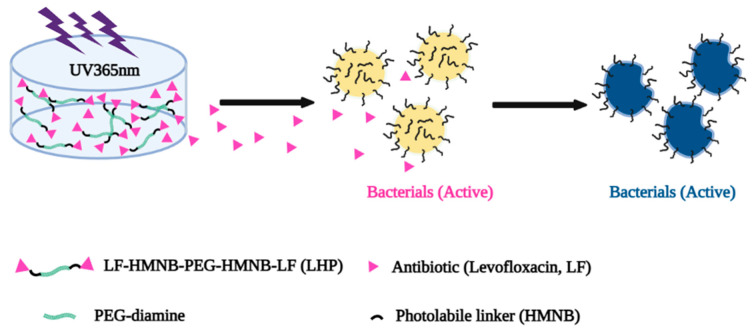
Schematic design of a photo-cleavable poly-prodrug (LHP)-loaded polyvinyl alcohol/sodium alginate (PVA/SA) wound dressing. Under UV irradiation at 365 nm, Levofloxacin (LF) can be cleaved from the LHP and released gradually from the wound dressing under UV irradiation [257]. This figure was drawn by “Untitled (biorender.com) (accessed on 20 April 2023)”.

**Figure 5 pharmaceutics-15-01829-f005:**
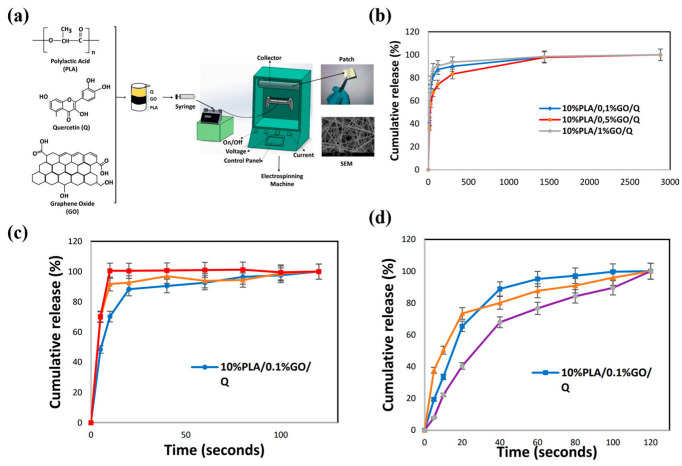
Schematic illustration of electrically triggered drug delivery from electrospun poly(lactic acid)/graphene oxide/quercetin fibrous scaffolds: (**a**) The idea, fabrication, and methods; (**b**) cumulative release graph of the drug-loaded microfiber scaffolds without electric stimulus; and cumulative release graphs under electric stimulus at two different frequency values: 10 Hz (**c**) and 50 Hz (**d**). Reprinted with permission from Ref. [263] (copyright 2021, Croitoru).

**Figure 6 pharmaceutics-15-01829-f006:**
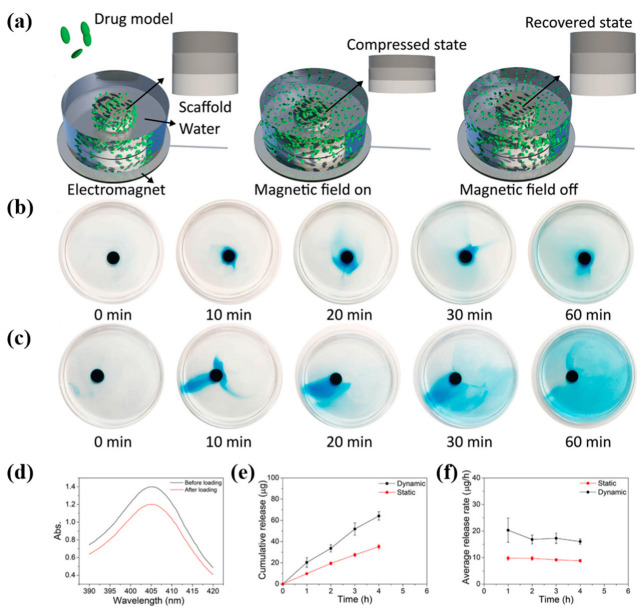
Magnetically responsive nanofibrous ceramic scaffolds for on-demand motility and drug delivery: (**a**) Schematic representation of the process of on-demand drug release from hydroxyapatite (HA)/Fe_3_O_4_ sandwich scaffolds induced by alternating magnetic field; images of methylene blue dye release from HA/Fe3O4 sandwich scaffolds without (**b**) and with (**c**) magnetic stimulation within 1 h; (**d**) UV–vis absorption spectra of the hemoglobin (Hb) solution before and after immersion of an HA/Fe_3_O_4_ scaffold; (**e**) cumulative release profiles of Hb from HA/Fe_3_O_4_ sandwich scaffolds subject to or without magnetic stimulation over 4 h; and (**f**) average Hb release rate of HA/Fe_3_O_4_ sandwich scaffolds subjected to or without magnetic stimulation at different time points over 4 h. Reprinted with permission from Ref. [268] (copyright 2022, Zhang).

**Table 1 pharmaceutics-15-01829-t001:** Polymers used for the preparation of nanofiber scaffolds.

			Pros	Cons	References
**Natural polymer**	Proteinaceous polymers	Collagen	Good biocompatibility Good biodegradability Possesses natural bioactivity	Poor customizability, Significant batch-to-batch variations, Inferior mechanical properties	[73]
		Fibrinogen			[74]
		Elastin			[75]
		Keratin			[76]
		Soy protein			[77]
		Silk protein			[78]
		Gelatin			[79]
		Peanut protein isolate			[80]
	Polysaccharide polymers	Chitosan			[81]
		Dextran			[82]
		Agarose			[83]
		Starch			[84]
		Sulfated alginates			[85]
		Alginate			[78]
		Iota carrageenan			[86]
		Kappa carrageenan			[87]
		Hyaluronic acid			[88]
		Bacterial cellulose(BC)			[89]
		Plant cellulose(PC)			[90]
		Cellulose acetate (CA)			[91]
		Xanthan gum			[92]
		polydeoxyribonucleotides (PDRN)			[93]
		Cashew gum(CG)			[94]
		Chondroitin sulfate			[95]
**Synthetic polymers**	Polylactic acid (PLA)		High customizability, High repeatability, Good mechanical properties	Poor biocompatibility, Difficult to degrade or rapid degradation, Lack of natural bioactivity	[96]
	Poly-L-lactic acid(PLLA)				[80]
	Polyglycolic acid (PGA)				[97]
	Poly(lactic-co-glycolic) acid (PLGA)				[98]
	ε-Polycaprolactone (PCL)				[99]
	Poly(L-lactic acid-co-ε-caprolactone) (PLCL)				[100]
	Polyurethane (PU)				[101]
	Poly(pyrogallol)				[102]
	Polyethylene glycol (PEG)				[103]
	Polyhydroxybutyrate (PHB)				[104]
	polyhydroxyalkanoate (PHA)				[105]
	Polyvinyl alcohol (PVA)				[94]
	Polyvinylpyrrolidone (PVP)				[106]
	Polysuccinimide (PSI)				[107]
	Poly(ethylene oxide) (PEO)				[108]
	Polysulfone				[109]
	Polythiophenen				[21]
	Aromatic polyimide				[110]
	Aramid				[111]
	Polyacrylonitrile(PAN)				[112]
	Poly(ether-ether-ketone)				[113]
	Polyimide				[114]

**Table 2 pharmaceutics-15-01829-t002:** Recent literature on the preparation of nanofibrous scaffolds after polymer mixing for skin-tissue engineering.

Scaffold Material	Additional Polymer	Bioactive Ingredients	Highlights	References
Alginate	PCL	Ag NPs, plasmid DNA encoding platelet-derived growth factor-B (PDGF-B), polyethyleneimine (PEI)	Highly absorbent alginate provides a moist environment for the wound, PCL increases cell adhesion, and the scaffold adsorbs drugs through electrostatic interactions.	[164]
	Silk Fibroin	*Lactobacillus casei*	*Lactobacillus casei*-loaded scaffolds introduce lactic acid with antimicrobial and wound-healing properties. In vitro, the cell-free supernatant of *Lactobacillus casei* inhibited the conversion of fibroblasts to myofibroblasts and attenuated endoplasmic reticulum stress.	[78]
	Gelatin	hydroxyl-rich silica nanoparticles	The incorporation of hydroxyl-rich silica nanoparticles into the sodium alginate/gelatin composite fiber greatly improved the hydrophilic, toughness, and axial shrinkage properties of the fiber. This fiber exhibits dynamic shrinkage behavior with humidity and can adapt to different wound shapes.	[165]
	Sodium Carboxymethyl Cellulose	-	Physical cross-linking by microwave treatment enhances the mechanical properties of the scaffold, thereby improving the diabetic wound-healing process and accelerating skin tissue regeneration.	[166]
	Microcrystalline cellulose(MCC), PVA	*Euphorbia humifusa* Willd. (EHW)	PVA and MCC enhance the mechanical properties of the fibers, while EHW enhances the antimicrobial and hemostatic properties of the dressing.	[167]
Cellulose	PVA	Carbon quantum dots (CQDs)- Fe_3_O_4_, rosemary extract (RE)	Carbon quantum dots (CQD)-Fe_3_O_4_ were introduced as a novel antibacterial agent, with which rosemary extract (RE) was complexed to reduce its cytotoxicity.	[168]
Chitosan	Gelatin, cellulose nanocrystal (GCCNC)	-	Slows down degradation and improves mechanical, enzymatic, and thermal stability.	[169]
	Silk fibroin	Nitrogen-doped carbon quantum dots, α-tricalcium phosphate	Good antibacterial and biocompatible properties against *Escherichia coli* and *Staphylococcus aureus*, promotes migration and proliferation, accelerates wound closure and re-epithelialization.	[170]
	HA, PVA	Temporin-Ra peptide	The peptide molecules are immobilized by electrostatic surface adsorption on a synthetic scaffold and are slowly released into the alkaline environment of the wound bed in vivo.	[81]
	PVA	Mupirocin, bupivacaine	Optimized water resistance and biodegradability of wound dressings.	[171]
	PEO	AgNPs, curcumin	Antibacterial and antiscarring properties.	[172]
	PEO	Kaolin	Adjustable mechanical properties, good biocompatibility, and hemostasis.	[173]
Hyaluronic acid	PVA	*Plantago major* Extract	Improves physicochemical and thermal properties and storage stability.	[174]
	PU	Propolis	Good biocompatibility, accelerated wound-healing process, and wound closure, improved dermal development and collagen deposition, and good antibacterial activity.	[175]
	Collagen	AgNPs, Gentamicin (GENT)	It is recommended to load AgNPs or gentamicin (GENT) alone, rather than commingling.	[176]
Starch	Hydroxypropyl methylcellulose (HPMC)	Zinc oxide nanoparticles (ZnO-NPs)	Excellent moisture absorption, water-vapor transmission rate, oxygen transmission rate, swelling capacity, and antibacterial activity.	[177]
	PVA, chitosan	-	Enhanced water resistance, optimized biodegradation rate, suitable mechanical properties in both dry and wet states, and excellent antibacterial activity against both Gram-negative and Gram-positive bacteria.	[178]
	Thermoplastic polyurethane (TPU)	-	Increases the water stability and mechanical properties of nanofibers.	[137]
	PU	AgNPs	Hydroxypropyl starch can increase the hydrophilicity of PU.	[179]
Silk fibroin	PVA, sodium alginate (SA), gelatin methacryloyl (GelMA)	-	Using a handheld electrospinning device, an aqueous solvent-based hydrogel–nanofiber composite structure is formed by photocross-linking after absorption of exudate.	[180]
	PCL	Lovastatin	Topographic cues perpendicular to the direction of tension and lovastatin act synergistically to inhibit mechanotransduction and fibrosis progression.	[181]
Collagen	Zein, PCL	ZnO NPs, *Aloe vera*	Suitable thermal stability and mechanical properties, adjustable water contact angle, and inhibitory activity on *Staphylococcus aureus* and *Escherichia coli*.	[182]
Gelatin	PCL	Adipose-derived stem cells (ADSCs)	Combined with collagen/alginate (Col/Alg) hydrogels to form a bilayer scaffold, ADSCs inoculated therein exhibited the best re-epithelialization, collagen organization, neovascular formation, and reduction in inflammation in the wound area.	[183]
	PCL	Starfish polydeoxyribonucleotides	Authors extracted polydeoxyribonucleotides (PDRN) from *Patiria pectinifera*, which possesses wound-healing activity.	[93]
	Graphene oxide (GO)	N-Acetyl Cysteine (NAC)	The NAC–GO-Gel stent provides greater mechanical properties and maintains NAC release than a single Gel stent, resulting in better cell proliferation and migration capabilities.	[184]
	PVA, PLGA	Thrombin (TMB), vancomycin (VCM)	Mixing PVA and Gel increases scaffold flexibility and reduces the probability of cytotoxicity, while PLGA adjunct helps to achieve prolonged and continuous drug delivery.	[98]
Rosmarinic acid (RA)	*Bletilla striata* polysaccharide (BSP), PVA, PLA	-	Good air permeability, flexibility, and biocompatibility; facilitates the proliferation and transformation of early wound macrophages; and downregulates MPO^+^ expression at the wound.	[96]
Cashew gum (CG)	PVA	-	Antibacterial activity against *Escherichia coli* and *Staphylococcus aureus* and has the ability to induce scarless wound healing.	[94]

## Data Availability

Not applicable.

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
