# Peer review of "Nanofiber Scaffolds as Drug Delivery Systems Promoting Wound Healing"

_pharmaceutics, 2023, doi:10.3390/pharmaceutics15071829_

Round 1

Reviewer 1 Report

Regarding this paper by title: Nanofiber Scaffold as a Drug Delivery System for Promoting Wound Healing: A Comprehensive Review of Recent Advances. this topic is good idea promising for researchers in the world. it's evident that authors have a lot of time into invested this review paper. But I have some comments for improvement this paper that should be applied before publishing.

1: please add novelty statement in abstract section. this issue is one of the important for readers. 

2: please add some section and figures related to this topic within this review paper. this work can be increasing quality your research.

3: please bolled point by point challenges ,limitaions ,advantage ,disadvantage .; this work can be making a significant progress to achieve your goals 

5: Please add more recent research about Drug Delivery System for Promoting Wound Healing into introduction

1:Bioactive Graphene Quantum Dots Based Polymer Composite for Biomedical Applications

2:Bioactive Agent-Loaded Electrospun Nanofiber Membranes for Accelerating Healing Process: A Review

3:Recent Advancements in Polythiophene-Based Materials and their Biomedical, Geno Sensor and DNA Detection

Regarding this paper by title: Nanofiber Scaffold as a Drug Delivery System for Promoting Wound Healing: A Comprehensive Review of Recent Advances. this topic is good idea promising for researchers in the world. it's evident that authors have a lot of time into invested this review paper. But I have some comments for improvement this paper that should be applied before publishing.

1: please add novelty statement in abstract section. this issue is one of the important for readers. 

2: please add some section and figures related to this topic within this review paper. this work can be increasing quality your research.

3: please bolled point by point challenges ,limitaions ,advantage ,disadvantage .; this work can be making a significant progress to achieve your goals 

5: Please add more recent research about Drug Delivery System for Promoting Wound Healing into introduction

1:Bioactive Graphene Quantum Dots Based Polymer Composite for Biomedical Applications

2:Bioactive Agent-Loaded Electrospun Nanofiber Membranes for Accelerating Healing Process: A Review

3:Recent Advancements in Polythiophene-Based Materials and their Biomedical, Geno Sensor and DNA Detection

Author Response

Dear Reviewer,
We appreciate your professional comments on our article, and as you were concerned that there were some issues that needed to be addressed, we have made extensive corrections to the previous manuscript based on your suggestions, as follows:

1: please add novelty statement in abstract section. this issue is one of the important for readers.

The author’s answer:

        We sincerely thank you for your valuable comments and we have rewritten the abstract section and hope that the changes will meet with your approval.

2: please add some section and figures related to this topic within this review paper. this work can be increasing quality your research.

The author’s answer:

        Thank you very much for your careful reading and professional advice, we have added 1 table and 5 diagrams to the manuscript, included a summary of the latest materials for the preparation of polymer blends as raw materials for nanofiber applications in skin tissue engineering in the tables section(L. 481-L. 503), and have included figures corresponding to typical references by stimulus type in the stimulus-responsive nanofiber scaffolds section, respectively. In addition, we have added "2.2.3 Multi-polymer blends" and "2.1.5. 3D printing technology" to the text, and in the discussion section, more expanded content has been added to the discussion section.

3: please bolled point by point challenges ,limitaions ,advantage ,disadvantage .; this work can be making a significant progress to achieve your goals

The author’s answer:

        We would like to thank you for your valuable suggestions. We apologise for the lack of logic and generality in our description of nanofibre preparation methods, which did not clearly highlight the advantages and disadvantages of each method. For this reason, we have included a summary of the challenges, limitations, advantages and disadvantages of each method in the five subsections 2.1.1 - 2.1.5.

4: Please add more recent research about Drug Delivery System for Promoting Wound Healing into introduction

1)Bioactive Graphene Quantum Dots Based Polymer Composite for Biomedical Applications

2)Bioactive Agent-Loaded Electrospun Nanofiber Membranes for Accelerating Healing Process: A Review

3)Recent Advancements in Polythiophene-Based Materials and their Biomedical, Geno Sensor and DNA Detection

The author’s answer:

       Thank you very much for your professional comments and we have added your suggested references and related content to the introduction at L. 312 [19]; L. 318 [20]; L. 326 [21].

5: Extensive editing of English language required

The author’s answer:

We apologize for any spelling errors caused by our carelessness and for any inconvenience caused by inaccurate language that you may have read. This manuscript has undergone MDPI's English editing service, all of which we hope will bring it up to the standards of the journal. Thank you very much for your valuable comments.

Reviewer 2 Report

Jiang et al. have published an interesting review article on nanofiber scaffolds as drug delivery systems for wound healing.

This review focuses on the fabrication methods of nanofiber biomaterials with drug delivery capability, the type of biomaterials, and the type of stimuli for controlled drug delivery in wound healing.

This review provides a very good summary of the explanation of nanofiber scaffolds in wound healing.

However, some of the improvements listed below should be considered before publication.

-Gradients and injectable biomaterials as advanced systems should also be included in the review.

-3D (bio)printing should be explained in the section "2.1 Manufacturing Processes for Nanofiber Scaffolds."

-I also recommend the inclusion of other important articles in the field of antibacterial and ph-responsive drug delivery biomaterials in wound healing.

Therefore, I recommend the following interesting articles/reviews as some examples of the above topics.

e.g., ACS Biomater. Sci. Eng. 2021, 7, 3, 1147–1158.

ACS Appl. Mater. Interfaces 2021, 13, 43755

Adv. Healthcare Mater. 2021, 2001341

Materials Science and Engineering: C, Volume 128, September 2021, 112264

Materials Today. 2020, 5:100051, DOI: https://doi.org/10.1016/j.mtadv.2019.100051

ACS Nano 2021, 15, 7, 12375–12387

Biomater. Sci., 2018,6, 340-349

Biomater Adv . 2022 Aug;139:212980. doi: 10.1016/j.bioadv.2022.212980.

ACS Appl. Bio Mater. 2022, 5, 6, 2726–2740

Author Response

Dear Reviewer,
We appreciate your professional comments on our article, and as you were concerned that there were some issues that needed to be addressed, we have made extensive corrections to the previous manuscript based on your suggestions, as follows:

  1. Gradients and injectable biomaterials as advanced systems should also be included in the review.

The author’s answer:

        Thank you for your valuable feedback. Based on your suggestions, we have included the section on gradient and injectable biomaterials as advanced materials in our manuscript, with injectable biomaterials described at L. 1662 and multi-gradient biomaterials discussed at L. 1653.

  1. 3D (bio)printing should be explained in the section "2.1 Manufacturing Processes for Nanofiber Scaffolds."

The author’s answer:

        Thank you for your valuable comments. As the reviewer pointed out, 3D (bio)printing is one of the important methods for the preparation of nanofibres, so we have added the section "2.1.5. 3D Printing technology" to the section "2.1 Manufacturing processes for nanofibre scaffolds" (at L. 267).

  1. I also recommend the inclusion of other important articles in the field of antibacterial and ph-responsive drug delivery biomaterials in wound healing.

e.g., ACS Biomater. Sci. Eng. 2021, 7, 3, 1147–1158.

ACS Appl. Mater. Interfaces 2021, 13, 43755

Adv. Healthcare Mater. 2021, 2001341

Materials Science and Engineering: C, Volume 128, September 2021, 112264

Materials Today. 2020, 5:100051, DOI: https://doi.org/10.1016/j.mtadv.2019.100051

ACS Nano 2021, 15, 7, 12375–12387

Biomater. Sci., 2018,6, 340-349

Biomater Adv . 2022 Aug;139:212980. doi: 10.1016/j.bioadv.2022.212980.

ACS Appl. Bio Mater. 2022, 5, 6, 2726–2740

The author’s answer:

        Your valuable comments are much appreciated and we have cited your recommended references in this manuscript with relevant additions, specifically: L. 3622 [285]; L. 3616 [283]; L. 3608 [282]; L. 3620 [284]; L. 3596 [280]; L. 980 [53]; L. 2097 [220]; L . 3597[281]; L. 3626[286].

Reviewer 3 Report

Journal Title: Pharmaceutics

Manuscript Title: Nanofiber Scaffold as a Drug Delivery System for Promoting Wound Healing: A Comprehensive Review of Recent Advances

Manuscript ID: pharmaceutics-2400907

Authors: Ziwei Jiang et al.

The major concern of the current review paper is its lack of novelty. Actually, a ”comprehensive” coverage of the chosen subject is quite daring. The literature survey reveals remarkable reviews on the subject (see for example https://doi.org/10.1016/j.biopha.2022.113996).

The way of presentation here is inappropriate. Just one table and one figure in the whole manuscript, which is claimed to be a review of the literature, is unacceptable. The authors leave the reader with the impression that they did not their best.

The content primarily consists of a systematic and logically arranged description of literature data. However, some of the details mentioned may not be relevant or accurate. For example, the discussion on chitosan is irrelevant as this natural polymer cannot produce free-standing electrospun membranes by itself - it must be combined with other polymers. Therefore, any such combination should be described.

Due to the extensive nature of the subject, it was expected that certain classes of polymers or materials would not be addressed. For instance, while natural-synthetic composites were frequently studied to combine their benefits and overcome some disadvantages as single matrices, no references were made regarding composites in Table 1.

Furthermore, there are several other synthetic polymers such as poly(ether-ether-ketone)s, polysulfones, and polyimides that could have been mentioned in the list. Further discussion could also include multicomponent matrices with nanoparticles, which are of interest from both academic and biomedical perspectives.

In conclusion, the structure of the content must be rearranged, the authors may choose a more targeted direction (for example, indexing only one technique – electrospinning or other).

The Abstract and Introduction sections must be improved!

Under the Section 3. Application of Nanofibrous Scaffolds in Wound Healing the listed 3.1, 3.2, 3.3… subsections are rather biological processes that accompanies the wound healing process. Must be checked.

Utilizing graphics and tables is essential!

The typo and grammar should be checked.

Journal Title: Pharmaceutics

Manuscript Title: Nanofiber Scaffold as a Drug Delivery System for Promoting Wound Healing: A Comprehensive Review of Recent Advances

Manuscript ID: pharmaceutics-2400907

Authors: Ziwei Jiang et al.

The major concern of the current review paper is its lack of novelty. Actually, a ”comprehensive” coverage of the chosen subject is quite daring. The literature survey reveals remarkable reviews on the subject (see for example https://doi.org/10.1016/j.biopha.2022.113996).

The way of presentation here is inappropriate. Just one table and one figure in the whole manuscript, which is claimed to be a review of the literature, is unacceptable. The authors leave the reader with the impression that they did not their best.

The content primarily consists of a systematic and logically arranged description of literature data. However, some of the details mentioned may not be relevant or accurate. For example, the discussion on chitosan is irrelevant as this natural polymer cannot produce free-standing electrospun membranes by itself - it must be combined with other polymers. Therefore, any such combination should be described.

Due to the extensive nature of the subject, it was expected that certain classes of polymers or materials would not be addressed. For instance, while natural-synthetic composites were frequently studied to combine their benefits and overcome some disadvantages as single matrices, no references were made regarding composites in Table 1.

Furthermore, there are several other synthetic polymers such as poly(ether-ether-ketone)s, polysulfones, and polyimides that could have been mentioned in the list. Further discussion could also include multicomponent matrices with nanoparticles, which are of interest from both academic and biomedical perspectives.

In conclusion, the structure of the content must be rearranged, the authors may choose a more targeted direction (for example, indexing only one technique – electrospinning or other).

The Abstract and Introduction sections must be improved!

Under the Section 3. Application of Nanofibrous Scaffolds in Wound Healing the listed 3.1, 3.2, 3.3… subsections are rather biological processes that accompanies the wound healing process. Must be checked.

Utilizing graphics and tables is essential!

The typo and grammar should be checked.

Author Response

Dear Reviewer,
We appreciate your professional comments on our article, and as you were concerned that there were some issues that needed to be addressed, we have made extensive corrections to the previous manuscript based on your suggestions, as follows:

  1. The major concern of the current review paper is its lack of novelty. Actually, a ”comprehensive” coverage of the chosen subject is quite daring.

The author’s answer:

        We sincerely thank you for your valuable comments. Our review did not make the novelty and importance clear in the abstract section of the original manuscript, and in view of this, we have strengthened the introduction section to emphasize the novelty. The novelty of this paper lies in (1) the systematic summarization of the preparation process of nanofiber scaffolds, including preparation methods, synthetic raw materials, and drug delivery methods, which can help readers to have a clear understanding of the choice of synthesis and preparation methods as well as the selection of raw materials for nanoscaffolds that are used in the field of skin tissue engineering. (2) The specific roles of drug-loaded nanofibers in wound healing are reviewed from different stages of wound healing, which can help readers to understand the target process of drug-loaded nanofiber intervention. (3) summarizes the principles and recent applications of stimulus-responsive nanofiber scaffold drug delivery systems and looks at the combination of multiple systems for controlled drug release. This will enable the reader to gain an understanding of the more complex drug delivery systems of nanofibers and the possible directions of material intelligence, and inspire more researchers to invest in related fields of research.

  1. Just one table and one figure in the whole manuscript, which is claimed to be a review of the literature, is unacceptable.

The author’s answer:

Thank you for your reminder that as a literature review, our previous manuscript did contain too few figures and tables. For this reason, we have included a summary of the latest materials for the preparation of polymer blends as raw materials for nanofiber applications in skin tissue engineering in the tables section(L. 481-L. 503), and have included figures corresponding to typical references by stimulus type in the stimulus-responsive nanofiber scaffolds section, respectively. Once again, we sincerely thank you for your suggestions.

  1. the discussion on chitosan is irrelevant as this natural polymer cannot produce free-standing electrospun membranes by itself - it must be combined with other polymers. Therefore, any such combination should be described.

The author’s answer:

        Thank you very much for your professional comments and we agree with your comment that chitosan on its own is not spinnable and must be combined with other polymers in its application as a feedstock, therefore we have added a note in this section of 2.2.3 (L. 467-L. 481) and listed in Table 2 other polymers that have been electrospun with chitosan in recent studies for applications in skin tissue engineering. However, as chitosan is one of the representative raw materials for natural polymers, we have not removed its discussion in the natural polymer raw materials, although chitosan is often not used in isolation.

  1. Due to the extensive nature of the subject, it was expected that certain classes of polymers or materials would not be addressed. For instance, while natural-synthetic composites were frequently studied to combine their benefits and overcome some disadvantages as single matrices, no references were made regarding composites in Table 1.

The author’s answer:

As you have noted, the breadth of the subject matter makes it difficult to list the full range of nanofibre feedstocks, but we are willing to do our best to retrieve relevant research in wound healing to round out this section. As far as possible, we have included in the new Table 2 a list of recent research in the field of skin tissue engineering where natural polymers have been blended with other polymers to improve mechanical properties and spinnability.

  1. Furthermore, there are several other synthetic polymers such as poly(ether-ether-ketone)s, polysulfones, and polyimides that could have been mentioned in the list.

The author’s answer:

Thank you very much for your professional comments and the additions you have helped us to make. Based on your suggestions we have added several more synthetic polymers to Table 1, including: polythiophenen, aromatic polyimide, aramid, polyacrylonitrile (PAN), poly(ether- ether-ketone), polyimide (L. 308-L. 314). Thank you again for your suggestions.

  1. Further discussion could also include multicomponent matrices with nanoparticles, which are of interest from both academic and biomedical perspectives.

The author’s answer:

Thank you very much for your suggestion and we couldn't agree with you more that the exploration of multi-component matrices for nanoparticles is significant from both an academic and biomedical point of view and we have added this part of the discussion to the manuscript at L. 1120.

  1. The Abstract and Introduction sections must be improved!

The author’s answer:

        We sincerely thank the reviewers for their careful reading and, in response to their comments, we have rewritten the abstract section and substantially revised the introduction section in the hope of improving it.

  1. Application of Nanofibrous Scaffolds in Wound Healing the listed 3.1, 3.2, 3.3… subsections are rather biological processes that accompanies the wound healing process. Must be checked.

The author’s answer:

        We sincerely thank the reviewers for their careful reading and valuable feedback, and apologise that our language was not clear and precise enough that we may have misunderstood the reader in this section. In sections 3.1 to 3.6, we have categorised the drug-loaded nanofibres that play a role in different processes of wound healing, using the wound healing process as a cue, and our focus in this section is on what role the drug-loaded nanofibres play in facilitating the specific processes of wound healing.

  1. Utilizing graphics and tables is essential!

The author’s answer:

        Thank you again for your reminder, we have added a table and five pictures which we hope will bring our articles more in line with the standards of your magazine.

  1. The typo and grammar should be checked.

The author’s answer:

We apologize for any spelling errors caused by our carelessness and for any inconvenience caused by inaccurate language that you may have read. This manuscript has undergone MDPI's English editing service, all of which we hope will bring it up to the standards of the journal. Thank you very much for your valuable comments.

  1. In conclusion, the structure of the content must be rearranged, the authors may choose a more targeted direction (for example, indexing only one technique – electrospinning or other).

The author’s answer:

        Thank you very much for your valuable comments and professional comments, in order to make the logic of the article more understandable to the reader, we have rewritten the summary section and improved most of the content in the introduction, and added figures and tables to the article to facilitate a clearer and more comprehensive understanding of the content of each part, in addition, we have added "2.2.3 Multi-polymer blends" and "2.1.5. 3D printing technology" makes the two parts of nanofiber preparation method and nanofiber synthesis raw material more complete. Our article aims to summarize the use of drug-loaded nanofiber scaffolds in skin tissue to provide more advanced candidates for wound dressings that can be used in clinical work in the future. At the same time, we add more discussion sections to elaborate on drug-loaded nanofiber materials currently applied to skin tissue engineering and discuss complex drug delivery systems that can be combined with nanoscaffolds.

Reviewer 4 Report

This review deals with the current status of nanofiber drug delivery systems. The authors describe the common methods for preparing nanofiber scaffolds and for combining drugs, as well as the polymers used for making nanofibers. In addition, the application of drug-loaded fiber scaffolds in the wound healing are discussed, and several methods by utilizing stimuli-responsive nanofiber drug delivery systems are exemplified. This review is well organized and documented, being worth publishing.

The authors are requested to revise their manuscript by considering the following points.

1) Section 2.1: In each of the sub-sections 2.1.1-2.1.4, the corresponding Figure 1 (a) –(d) must be well described. Otherwise, the detail of each figure and the characteristics of each method cannot be understood comparably.

2) L. 284: “sericin 1 and sericin 2”. Show some reference for readers to know their difference in structure.

3) L. 294: “(ROP) [127, 128].” These references are inappropriate for the description here.

4) 2.2.3 Coating method: The difference between Physical adsorption method and Coating method is unclear. Better description is needed for the differentiation.

5) It should be necessary to show several real applications of the nanofiber-drug formulations, if any. If no application has been done yet, both the reason and the future prospect should be shown in the conclusive remarks.

Author Response

Dear Reviewer,
We appreciate your professional comments on our article, and as you were concerned that there were some issues that needed to be addressed, we have made extensive corrections to the previous manuscript based on your suggestions, as follows:

1) Section 2.1: In each of the sub-sections 2.1.1-2.1.4, the corresponding Figure 1 (a) –(d) must be well described. Otherwise, the detail of each figure and the characteristics of each method cannot be understood comparably.

The author’s answer:

We sincerely thank the reviewers for their careful reading and valuable feedback, and we have added more detailed annotations to Figs. 1(a)-(d). In addition, when using the same method for preparing nanofibres, the variability introduced by different components and ratios of materials as raw materials makes the specific processes in the preparation process of the cited literature not universal, so we have not, for the time being, included Figs. 1(a)-(d) of The specific preparation processes involved in the experiments are described in detail in each of the subsections 2.1.1-2.1.4, and instead we have retained the initial generalised step-by-step descriptions. Thank you again for your suggestion and we are more than willing to make further changes as suggested by the reviewers.

2) L. 284: “sericin 1 and sericin 2”. Show some reference for readers to know their difference in structure.

The author’s answer:

        We sincerely thank you for your valuable comments and we have carefully checked the references and added more references to "sericin 1 and sericin 2" in the revised version of L. 394-L. 408.

3) L. 294: “(ROP) [127, 128].” These references are inappropriate for the description here.

The author’s answer:

Thank you very much for your careful reading, we apologise for our carelessness and we have replaced the appropriate reference in L. 414 and thank you for the correction.

4) 2.2.3 Coating method: The difference between Physical adsorption method and Coating method is unclear. Better description is needed for the differentiation.

The author’s answer:

        Thank you very much for the valuable feedback from the reviewers, we were indeed unclear in this section and we have added a description of this section at L. 556 to distinguish between the physical adsorption method and the coating method.

5) It should be necessary to show several real applications of the nanofiber-drug formulations, if any. If no application has been done yet, both the reason and the future prospect should be shown in the conclusive remarks.

The author’s answer:

        Many thanks to the reviewers for their expert advice, which we think is a good one, and we have described in detail the clinical applications and future prospects of drug-laden nanofibres at manuscript L. 1709.

Round 2

Reviewer 3 Report

Journal Title: Pharmaceutics

Manuscript Title: Nanofiber Scaffold as a Drug Delivery System for Promoting Wound Healing: A Comprehensive Review of Recent Advances

Manuscript ID: pharmaceutics-2400907, 1st Revision

Authors: Ziwei Jiang et al.

The authors have addressed main questions raised upon first evaluation. The paper become more significant in its content.

-line 33, check the meaning of ”do not have possess”

-the legend from Fig 1, the whole text there should be transferred to text, and the legend should be shorter pointing out the letters referring the images shown. Also, the e) is missing (line 114). The way of typing references is inconsistent (.[23], .[24] etc.), the dots should be erased.

-figure 2 is copy paste from ref 261. Probably the authors need copyright authorization from the initial publisher.

-check the line 773. The section 2.4 is misspelled

-figure 3 is copy paste from ref 264. Probably the authors need copyright authorization from the initial publisher.

-same for all figures introduced in the revised version

Author Response

Dear Reviewers:

Thank you very much for your approval of our manuscript and for giving us the opportunity to make minor revisions. Thank you for taking your valuable time to review our manuscript and suggest changes, we have made the changes you pointed out and the point-to-point review responses are as follows:

-line 33, check the meaning of " do not have possess "
The author's response:

We apologize for our carelessness; we have changed this statement to "do not possess" and thank you very much for pointing out our mistake.

-check the line 773. The section 2.4 is misspelled

The author's response:

Perhaps due to formatting changes made by the editor, or inconsistencies in word versions, etc., what you describe is not on line 773 in the version of the manuscript we received. We presume that we may had a spelling problem and a formatting problem with subheading 4.2 on line 641 in the version we saw, so we have made changes to it.

-the legend from Fig 1the whole text there should be transferred to text and the legend should be shorter pointing out the letters referring the images shown.
The author's response:

Thanks to your professional advice, we have changed the notes to Figure 1 to make it more concise and the relevant details of Figures 1(a)-(e) have been added to the corresponding sections in the text.

-Also, the e) is missing(line 114). 
The author's response:

It is possible that due to issues such as formatting changes made by the editor, or inconsistent word versions, in the version of the manuscript we received, (e) should have appeared on line 100 but was missing. We checked our last revised submission of the manuscript and found that there was no missing figure 1 marker (e), probably due to an inadvertent deletion caused by formatting changes made by the editor, and we have included this marker in the latest version. We are very grateful for your careful review and professional advice to help us make the manuscript more rigorous.

-The way of typing references is inconsistent (.[23], [24] etc.), the dots should beerased.
The author's response:

Thank you very much for your careful checking, we are very sorry for our carelessness, our failure to check properly led to inconsistencies in the placement of references in relation to punctuation, again we apologise for our error and we have aligned the references in relation to punctuation.

-figure 2 is copy paste from ref 261. Probably the authors need copyright authorizationfrom the initial publisher.

-figure 3 is copy paste from ref 264. Probably the authors need copyright authorizationfrom the initial publisher.

-same for all figures introduced in the revised version.

The author's response:

We have sent online copyright applications or emails to the original publisher or author of each figure and proof that a copyright application has been obtained is shown below, with the authors of Figure 1(b)[24], Figure 5[269], and Figure 6[274] we sent emails requesting copyright permission and have not yet received a response.
